# The Phylum Bryozoa: From Biology to Biomedical Potential

**DOI:** 10.3390/md18040200

**Published:** 2020-04-09

**Authors:** Maria Letizia Ciavatta, Florence Lefranc, Leandro M. Vieira, Robert Kiss, Marianna Carbone, Willem A. L. van Otterlo, Nicole B. Lopanik, Andrea Waeschenbach

**Affiliations:** 1Consiglio Nazionale delle Ricerche (CNR), Istituto di Chimica Biomolecolare (ICB), Via Campi Flegrei 34, 80078 Pozzuoli, Italy; lciavatta@icb.cnr.it (M.L.C.); mcarbone@icb.cnr.it (M.C.); 2Service de Neurochirurgie, Hôpital Erasme, Université Libre de Bruxelles (ULB), 1070 Brussels, Belgium; 3Departamento de Zoologia, Centro de Biociências, Universidade Federal de Pernambuco, Recife, PE 50670-901, Brazil; leandromanzoni@hotmail.com; 4Retired – formerly at the Fonds National de la Recherche Scientifique (FRS-FNRS), 1000 Brussels, Belgium; rkiss2012@gmail.com; 5Department of Chemistry and Polymer Science, University of Stellenbosch, Private Bag X1, Matieland 7602, South Africa; wvo@sun.ac.za; 6School of Earth and Atmospheric Sciences, School of Biological Sciences, Georgia Institute of Technology, Atlanta, GA 30332, USA; nicole.lopanik@eas.gatech.edu; 7Life Sciences, Natural History Museum, Cromwell Road, London SW7 5BD, UK

**Keywords:** Alzheimer, anticancer, antiparasitic, antiviral, bryozoan, chemoecology, phylogeny

## Abstract

Less than one percent of marine natural products characterized since 1963 have been obtained from the phylum Bryozoa which, therefore, still represents a huge reservoir for the discovery of bioactive metabolites with its ~6000 described species. The current review is designed to highlight how bryozoans use sophisticated chemical defenses against their numerous predators and competitors, and which can be harbored for medicinal uses. This review collates all currently available chemoecological data about bryozoans and lists potential applications/benefits for human health. The core of the current review relates to the potential of bryozoan metabolites in human diseases with particular attention to viral, brain, and parasitic diseases. It additionally weighs the pros and cons of total syntheses of some bryozoan metabolites versus the synthesis of non-natural analogues, and explores the hopes put into the development of biotechnological approaches to provide sustainable amounts of bryozoan metabolites without harming the natural environment.

## 1. Introduction

Natural products have long been employed to ameliorate the quality of human life not only as nutrition but also as fragrances, pigments, insecticides, and medical drugs [1]. Of the 175 anticancer small-molecule drugs approved during the period 1940–2014, 75% are natural products or products derived from natural compounds [2,3]. Thus, many pharmaceutical groups remain interested in drug discovery from natural sources [4,5]. Although most of today’s natural product-derived drugs are of terrestrial origin, marine organisms have been revealed to be a huge reservoir of innovative medical drugs [6,7,8]. Within marine invertebrates, an important source of bioactive compounds, which remains only partially explored, is the phylum Bryozoa. Currently, more than 230 compounds have been isolated from 26 bryozoan species ([9] and previous reviews in this series). As this is only a minute proportion of the ~6000 currently described species, bryozoans represent an enormous source for the discovery of novel compounds. This review aims to provide a holistic, multidisciplinary account of the current state of affairs on this topic. It outlines aspects about general biology and evolutionary history of bryozoans and illustrates how bryozoans with the help of their metabolites shape chemoecologial interactions in the wild (Section 2). As anticancer activities of bryozoan metabolites have been recently reviewed by several authors [10,11,12], this review summarizes how selected compounds from bryozoans may complement existing treatments while outlining the shortcomings of current cancer treatments, as well as the problems and mechanisms associated with metastatic growth (Section 3 and Section 4). Compounds holding promise to treat Alzheimer’s disease, post-stroke damage, and Parkinson’s disease are treated in Section 5. Antiviral properties of bryozoan compounds are explored in Section 6, whereas antiparasitoidal activities are outlined in Section 7. This review also summarizes the limitations of harvesting wild bryozoans, culturing them and producing compounds heterologously (Section 8). Research into partial and total chemical synthesis as well as biosynthesis of bryozoan metabolites are discussed in Section 9. 

## 2. Bryozoans

### 2.1. General Biology

Bryozoa (also known as Ectoprocta, Polyzoa or sea mats or moss animals) are aquatic, mostly sessile colonial animals that consist of small modules called zooids. Feeding zooids typically consist of a calcified body wall and a soft-bodied part called polypide; the polypide consists of a ciliated tentacle crown (lophophore), a gut, and associated musculature and nerves. All bryozoans are suspension feeders, which means their lophophores capture organic particles out of the water column. Bryozoans comprise a comparatively poorly studied group despite their diversity (~6000 described extant species) [13]). They are ecologically important suspension feeders found in freshwater, brackish, and marine environments. Not only do they provide food for their predators (e.g., nudibranchs, sea spiders) but they also provide habitats for other animals such as small crustaceans, juvenile mussels, nematodes, entoprocts, etc.

Bryozoans typically form encrusting, massive, or erect colonies on hard natural and man-made substrates, but they can also be found living on sea weeds, sand grains, rooted in soft deep-sea sediments [14], or as free-living discs (e.g., O’Dea [15]). Amongst the more common growth forms are encrusting (e.g., *Cryptosula*, *Tegella*, and *Watersipora*), palmate (e.g., *Pentapora*), foliose (e.g., *Chartella*, *Flustra*, and *Sessibugula*), articulate (e.g., *Pterocella*, *Paracribricellina*), branched (e.g., *Myriapora*), arborescent (e.g., *Bugula*) and fenestrate (e.g., *Reteporella*); Figure 1 provides illustrations of some of these growth forms.

Most bryozoans exhibit a certain amount of polymorphism, meaning that zooids can be morphologically and functionally different. Typical functions of different zooid types are feeding, reproduction, and defense against micropredators and epizoites. 

In terms of reproduction, most bryozoans are considered to be hermaphroditic. Their colonies display either zooidal hermaphroditism or zooidal gonochorism (male and female zooids). The life cycle typically includes both sexual and asexual reproduction (Figure 2). It is assumed that self-fertilization in bryozoans only occurs when cross-fertilization is not possible [16]. Each new colony is initiated by a sexually produced planktonic larva which settles and metamorphoses into the founder zooid (ancestrula) which, in turn, buds other zooids by asexual reproduction (Figure 2). Most bryozoan species are brooders; their embryos typically develop into short-lived non-feeding larvae in special brood chambers called ovicells. However, there are some non-brooding species that produce planktotrophic larvae that remain in the plankton for longer periods of time. Reproductive seasonality and cycles of growth are variable among bryozoan species. The zooidal life span may range from one to a few weeks, whereas the life span of a colony ranges from a few weeks to twelve years [17]. The larvae may be produced continuously throughout the year, as in those species growing on ephemerous substrates, or they may have distinct sexual reproduction seasons.

### 2.2. Phylogeny

Work on plants has demonstrated that species of medicinal use tend to cluster on phylogenetic trees, and phylogenies have been shown to be useful predictive tools to guide the search of novel bioactive compounds [18,19,20]. For this reason, we felt it important to include a section on the current state of affairs of bryozoan phylogeny. Bryozoans are lophotrochozoan animals which are composed of three classes: Phylactolaemata, Stenolaemata, and Gymnolaemata, the latter comprising the orders Ctenostomatida and Cheilostomatida [21,22]. The least diverse of these is the Phylactolaemata, which comprises ~86 species, all of which are uncalcified and live in freshwater habitats [13]. The remaining species live either exclusively (Stenolaemata) or mostly (Gymnolaemata) in marine habitats, and most produce a calcium carbonate skeleton.

The cartoon in Figure 3 summarizes the current consensus concerning within-bryozoan interrelationships. Bryozoan terminals are given as higher level classification as derived from the World Register of Marine Species database (http://www.marinespecies.org [23]) (Phylactolaemata: families; Stenolaemata: families or Clades A–C (for clade details, see Waeschenbach et al. [24]); Ctenostomatida: superfamilies; Cheilostomatida: suborders (Malacostegina, Inovicellina, Scrupariina) or superfamilies). In cases where higher taxonomic groups were not monophyletic, genera representing those lineages are given.

The Phylactolaemata is the earliest diverging bryozoan lineage, and the Stenolaemata, represented by the only surviving order, the Cyclostomatida, forms the sister group to the Gymnolaemata [25,26,27]. Within the Gymnolaemata, the Ctenostomatida is paraphyletic to the inclusion of the Cheilostomatida [27] However, much ambiguity remains regarding the interrelationships within the Phylactolaemata, Ctenostomatida and Cheilostomatida.

Regarding the Phylactolaemata, Figure 3 summarizes the topology as found by Waeschenbach et al. [27]. Two phylactolaemate species have so far been examined for their bioactive compounds: *Pectinatella magnifica* and *Hyalinella punctata*. Although native to North America, *P. magnifica* is an invasive species which has extended its distribution throughout Europe and has also been recorded in Japan, Korea (see Balounová et al. [28]) and, most recently, China [29]. *Hyalinella punctata* nests within the genus *Plumatella* [30,31,32,33]. In terms of their bioactive compounds, both *H. punctata* [34] and *P. magnifica* [35] contain antimicrobial compounds, and a recent study on *P. magnifica* demonstrated the presence of cytotoxic activity (LD_50_ < 100 µg/mL) in various organic extracts (hexane, CHCl_3_, ethyl acetate, MeOH) with the exception of water extract [36]. Furthermore, methanolic extracts from *H. punctata* were shown to display immunomodulatory and anticancer activity in vitro [37], and water extracts from the same species contain antioxidants that display anti-carbon dioxide anion radical activity [38]. *Hyalinella punctata* also contains antioxidant volatiles [35].

A key outcome from molecular phylogenetic studies is that the current classification, which is based on morphological characters, is often not supported by molecular data [39]. This is particularly true for the Cyclostomatida, and phylogenetic reconstructions based on molecular data have raised serious implications for the classification of this group [24,40,41]. Two cyclostome species have so far been analyzed for bioactive compounds: *Heteropora alaskensis* and *Diaperoecia californica* [42]. Both genera are close relatives of each other within Clade C.

Within the Gymnolaemata, the Ctenostomatida were resolved as paraphyletic to the inclusion of the Cheilostomatida [27]. With regard to within superfamily interrelationships, the Vesicularioidea are the best-studied ctenostome group [43,44]. This is echoed by the number of species examined for bioactive compounds. Six of the eight ctenostome species examined for bioactive compounds are from the vesicularid genus *Amathia* (*A. alternata*, *A. convoluta*, *A. pinnata*, *A. tortuosa*, *A. wilsoni*, *A. verticillata*). The remaining two species are from the early diverging genus *Alcyonidium* (*A. diaphanum*, *A. gelatinosum*) [27].

Regarding the Cheilostomatida, two studies have tackled their overall phylogenetic relationships in recent years [27,39]. The suborder Malacostegina was paraphyletic in Waeschenbach et al. [27] to the inclusion of the suborders Scrupariina, Inovicellina, and Flustrina. The two malacostegan species which have so far been analyzed for novel compounds, *Biflustra grandicella* [45] and *B. perfragilis* [46], have not been put into a phylogenetic context, but considering they belong to the same family as *Membranipora*, they were tagged to the lineage represented by *Membranipora* (Figure 3). Regarding the interrelationships of the remaining cheilostomes, much uncertainty remains regarding the placement of the cribrimorphs, which are represented in Figure 3 by superfamilies Cribrilinoidea and Catenicelloidea. Cribrimorphs are likely non-monophyletic (see Gordon [47]) and, thus, dense taxon sampling of this group is highly recommended for future phylogenetic studies. Three cribrimorph species belonging to two abovementioned superfamilies have so far been analyzed for bioactive compounds: *Euthyroides episcopalis* [48,49], *Pterocella vesiculosa* [50,51,52] and *Paracribricellina cribraria* [53,54].

The cheilostome superfamilies that have been studied most intensively for their bioactive compounds are the Flustroidea, Buguloidea, and Calloporoidea. All three form a monophyletic clade together with the Hippothooidea (Figure 3). Thirteen species have so far been analyzed for bioactive compounds in this group: *Flustra foliacea*, *Securiflustra securifrons*, *Chartella papyracea*, *Hincksinoflustra denticulata*, *Tegella* cf. *spitzbergensis*, *Bugula neritina*, *Virididentula dentata*, *Bugulina flabellata*, *Caulibugula inermis*, *Tricellaria ternata*, *Dendrobeania murrayana*, and *Sessibugula translucens* [10,12].

Within the most derived clade in Figure 3, eight species have so far been examined for bioactive compounds: *Aspidostoma giganteum*, *Myriapora truncata*, *Primavelans insculpta*, *Phidolopora pacifica*, *Watersipora cucullata*, *Watersipora subtorquata*, *Pentapora fascialis*, and *Cryptosula pallasiana* [10,12]. 

In view of the topic of the present review, a better-sampled future molecular phylogeny ought to guide future efforts in discovering novel bioactive compounds either by targeting close relatives of taxa which have already yielded promising compounds or by targeting distantly related taxa to discover a greater diversity of compounds.

### 2.3. Chemoecology

As many natural compounds of medical importance are helpful for defense and/or spatial competition in their source organisms, this section reviews the ecological functions of bioactive compounds found in bryozoans. A decade ago, Porter and colleagues [55] published an excellent review documenting the various natural products isolated from bryozoans, detailing available pharmaceutical activities of these compounds as well as their proposed ecological roles including antimicrobial, antifouling, allelopathic, and antipredator activities.

The best-studied species in this context is the widespread fouling species *Bugula neritina* (Figure 1A). Rigorous ecological studies by Lindquist and Hay [56,57] demonstrated that *B. neritina* larvae were defended from both vertebrate and invertebrate predators. Curiously, adult colonies were much less deterrent, suggesting that the chemical defense was concentrated in the vulnerable larvae [56,57]. Further ecological studies demonstrated that *B. neritina* larvae are, in fact, defended from predators by bioactive compounds called bryostatins [58,59]. Bryostatins (e.g., bryostatin-1, (**1**), Figure 4) are complex polyketides whose anticancer activity was first discovered in 1970 [60]. *Bugula neritina* larvae that were cured of the symbiont by antibiotic treatment were significantly less deterrent than control larvae, indicating that a microbial symbiont is responsible for producing the deterrent bryostatins [58]. These results corroborated evidence from earlier morphological studies which found bacterial endosymbiont within both adult and larval tissues of *B. neritina* [61,62]. These findings were later confirmed by the Haygood group who discovered that a γ-proteobacterium symbiont named “*Candidatus* Endobugula sertula” likely produced the bryostatins found in *B. neritina* [63,64,65]. Fluorescent *in situ* hybridizations (FISH) showed “*Ca.* E. sertula” cells were associated with the ovicells, suggesting that the symbiont in the adult may produce bryostatins for the larvae [66]. This tritrophic interaction was the first in the marine environment to demonstrate that a microbial symbiont produces a compound that protects its host from predators.

Bryozoan host DNA sequences indicated that *B. neritina* comprises a complex of sibling species [64,67], termed Type S (shallow, <9 m), Type D (deep, >9 m), and Type N (North Atlantic). Type S is found in tropical, subtropical, and temperate waters, spread presumably by anthropogenic transport by fouling ship hulls [67,68]. By contrast, Type D is only found in California [64,67], whereas Type N is mostly found at locations around the US North Atlantic coast [69], but has also been found in California and Australia [67]. Types S and D host different strains of “*Ca.* E. sertula” along with different bryostatins [64], whereas Type N was originally shown not to possess the symbiont or deterrent bryostatins [69]. It was hypothesized that this is because colonies at higher latitudes face less predation pressure, as suggested by canonical biogeographic theory [70]. However, subsequent studies showed that Type N animals were shown to not be restricted to higher latitudes, and that Type S, which had always been shown to be symbiotic, was not confined to lower latitudes [67,71]. Furthermore, Type S animals which had been found at high latitudes had lost their symbionts, whereas most Type N colonies found at low latitudes had acquired the symbiont, indicating that the symbiont, and not the host, appeared to be more restricted by biogeography [71]. The symbionts in the Type N colonies are the same strain as that in the Type S, which differ from the strain in the Type D colonies. As the Type N and S hosts are genetically different, this suggests that, contrary to all previous evidence, the symbiont may be acquired from the environment [71].

Pettit et al. [72] concluded that *B. neritina* may use symbiotic associations as sources of defensive and/or offensive substances and suggested that their typically efficient colonization success may be due to their microbially-produced natural products.

Concerning other bugulid species, Lim and Haygood [73] showed that *Bugulina simplex* hosts a symbiont closely related to “*Ca.* E. sertula”, “*Candidatus* Endobugula glebosa”, which also produces compounds with bryostatin-like activity. Interestingly, they showed that two other species, *Bugulina turbinata* and *Crisularia pacifica*, possess a similar symbiont, which does not appear to be producing bryostatin-like compounds. This led the authors to suggest that the symbionts in these two species may have lost the genes for biosynthesizing bryostatin as the larvae are not as large and apparent as those of *Bugula neritina* and *Bugulina simplex*.

*Virididentula dentata* (previously known as *Bugula dentata*) is another bryozoan with a potential natural product producing microbial associate [74]. As with *B. neritina*, *V. dentata* comprises a widespread complex of sibling species [75], and further investigations may indicate similar patterns concerning the presence of symbionts as those known for *B. neritina*. Briefly, a blue tetrapyrrole pigment (**2**, Figure 5) was discovered in colonies of *V. dentata* [76] and was also similarly found in an Australian colonial ascidian [77]. As this pigment had previously been described as a product of a mutant strain of a bacterium [78], it was assumed that the true source of the compound isolated from both the bryozoan and ascidian is an associated bacterium. 

A symbiotic origin could be also postulated for tambjamines (**3a**–**3n**, Figure 5), a family of 4-methoxypyrrolic alkaloids which were found in different species of bryozoans (*V. dentata*, *Bugula longissimi*, *Sessibugula translucens*) [79,80,81,82,83] as well as in marine ascidians of the genus *Atapozoa* [84]. Tambjamines have been used to assess the trophic relationship between some bryozoan species and their predators, the mollusks *Tambja ceutae*, *T. stegosauriformis*, *T. eliora*, *T. abdere*, and *Roboastra tigris*, in which tambjamines were also found [80,81]. 

The ctenostome bryozoan genus *Amathia* is also a rich source of bioactive natural products, some with documented ecological activities such as feeding deterrence and antimicrobial activity [55]. Volutamides (**4a–4e**, Figure 6), which are present in the Atlantic bryozoan *Amathia convoluta*, deter feeding by potential predators and are toxic toward larvae of a co-occurring hydroid [85].

*Amathia wilsoni*, a large and common bryozoan occurring in Tasmanian coastal waters, contains a series of enamide alkaloids called amathamides A–G (**5a**–**5g**, Figure 7) [86,87,88]. Amathamide C (**5c**) has been produced by *Amathia convoluta* whereas *Amathia pinnata* contained amathamide G (**5g**, Figure 7) [89]. Blackman and Fu [90] isolated from *Amathia wilsoni* a β-phenylethylamine compound (**6**, Figure 7), which could be a biosynthetic precursor of amathamides. Total syntheses of amathamide A (**5a**) [91], B (**5b**) [91], D (**5d**) [92], and F (**5f**) [93] have already been performed and have led to the revision of the structures of amathamides D and F. Amathamides content was shown to vary between collection sites [87] and between different parts of the colony (high in exposed tips versus undetectable at the colony base) [86]. Thus, Walls et al. [86] suggested that amathamides could be involved in a chemical antipredator defense system in *A. wilsoni*. In a further study, Walls et al. [94] showed that amathamides are closely associated with a specific morphological bacterial type that is present on the surface of *A. wilsoni* and cautiously hypothesized that amathamides may be of bacterial origin. In line with these findings, Sherwood et al. [95] showed that amathamide C (**5c**, Figure 7) is able to deter feeding by fishes and could, therefore, serve as a chemical defense in *A. wilsoni*. In addition, Sherwood et al. [95] analyzed 34 epizoic species that colonize *A. wilsoni* and only one, the sea spider *Stylopallene longicauda*, which was present at high average densities, contained amathamides in concentrated amounts, thus, amathamides may also function as chemical defenses for *S. longicauda*. Another family of compounds from the genus *Amathia* are the convolutamines A–J (**7a**–**7j**, Figure 8) [96,97,98,99,100]. In particular, convolutamine F was shown to inhibit the cell division of fertilized sea urchin eggs [97].

Studies of ecological interactions in polar regions have provided important leads for future studies. Avila and colleagues examined the chemical ecology of Antarctic bryozoans [101,102,103,104]. Extracts from 11 out of an examined 17 species were deterrent to the omnivorous sea star *Odontaster validus*, and 13 out of 13 were repellent to the amphipod *Cheirimedon femoratus*, indicating a high percentage of species with chemical defenses [101,104]. They subsequently demonstrated that while none of the extracts from the 13 bryozoan species were cytotoxic towards sea urchin embryos, they were toxic for sea urchin sperm [102]. Further, substrate preferences of *C. femoratus* were negatively affected by extracts from 10 bryozoan species, indicating that chemical defenses are common in Antarctic bryozoans.

One question that arises when discussing a bioactive natural product, especially one that is produced by a microbial associate, is how the host has adapted to the presence of these compounds, especially when the natural product target is a eukaryotic protein. In *B. neritina*, this was investigated by comparing untreated control colonies with those that had their symbionts reduced with antibiotics [105]. Symbiont-reduced colonies had significantly fewer ovicells than the control colonies, indicating a reduced fecundity compared to symbiotic colonies. Furthermore, the expression of conventional protein kinase Cs (PKCs), which are signaling molecules that control cellular events that are activated by bryostatins, was different in the two colony types. However, expression of PKCs unaffected by bryostatins was similar between the two colony types, suggesting that the presence of the bryostatin-producing symbionts affects PKC expression in the host. Analysis of the *Bugula neritina* transcriptome led to the identification of five PKC isoforms which, in conjunction with bryostatins, are hypothesized to play a role in reproduction in *B. neritina* [105]. A detailed investigation of the morphology of female zooids from colonies with and without the symbiont did not reveal any anatomical differences that could account for the variation in overall fecundity [106]. In addition, expression levels of genes thought to be involved in invertebrate reproduction were similar in zooids from the two colony types. These data suggest that the symbiont or symbiont-produced bryostatins do not affect female anatomical structures or functions in the zooids, but instead may influence early differentiation of female germinal cells that may account for the observed differences in colony fecundity [106]. It is also unclear if these processes are influenced via PKC activation (or the lack thereof), but PKCs have been implicated in various reproductive processes in other invertebrates [107,108,109,110].

## 3. Bryozoans as Sources of Novel Anticancer Drugs

Bryozoan metabolites exhibiting in vitro anticancer properties were comprehensively and recently reviewed by Pejin et al. [111], Wu et al. [11], Tian et al. [10], and Figuerola and Avila [12]. Bioactive bryozoan metabolites were also discussed by Skropeta and Wei [112] reviewing advances in deep-sea natural product research, as well as by Hegazy et al. [113] reviewing biomedical leads from Red Sea marine invertebrates. In this section, we outline the effectiveness of selected bryozoan-derived metabolites to combat cancer, considering some of the major causes and mechanisms of cancer chemoresistance.

### 3.1. Cancer Stem Cells 

Cell populations of many cancers are thought to be sustained by cancer stem cells (CSCs), causing therapeutic refractoriness and dormant behavior [114,115,116]. In the past, CSCs were also referred to as tumor-initiating cells [115], but more recent evidence suggests that CSCs and tumor-initiating cells are different subsets of cell populations (see Valent et al. [117] for precise definitions and terminology of the various types of CSC types). Within bryozoans, the secondary metabolite bryostatin-1 (**1**, Figure 4) markedly potentiates the anticancer effects of imatinib mesylate (Gleevec^®^) in chronic myeloid leukemia (CML) stem cell elimination [118]. 

### 3.2. Multidrug Resistance (MDR) Phenotype

The MDR phenotype of cancer cells is caused by the activation of efflux pumps, mainly the ATP-binding cassette (ABC) transporters [119], of which there are seven subfamilies (ABCA to G) with 48 members [120]. In order to remain active against MDR phenotype cancer cells, it is crucial for novel anticancer compounds to not be a substrate for the ABC transporters. There are a number of strategies that can overcome MDR mechanisms, such as MDR modulators or chemosensitizers, multifunctional nanocarriers, and RNA interference (RNAi) therapy (reviewed by Saraswathy and Gong [121]). Many marine-derived compounds are capable of reversing or circumventing the MDR phenotype in cancer cells [122,123]. One of these compounds is bryostatin-1 (**1**, Figure 4) [122]. Sztiller-Sikorska [124] showed that bryostatin-1 is able to kill ABCB5 (ATP-binding cassette, sub-family B, member 5)-positive melanoma stem-like cells. Bryostatin-1 also downregulates MDR1 gene expression [125].

It was then experimentally demonstrated that i) bryostatin-1 is useful when combined with cytotoxic agents not only in vitro but also in vivo [125,126,127,128,129,130,131], and ii) bryostatin-1 downregulates MDR1 gene expression [125]. The fact that bryostatin-1 seems to display higher anticancer activity when combined with cytotoxic drugs could relate, at least partly, to the fact that bryostatin-1 is able to increase the sensitivity of cancer cells to pro-apoptotic stimuli [132,133]. All these experimental data prompted several oncologists to test bryostatin-1 in combined therapies against, for example, aggressive non-Hodgkin lymphomas with vincristine [134], in pancreatic cancer with paclitaxel [135], and in recurrent or persistent epithelial ovarian cancer with cisplatin [136], but in non-selected groups of cancer patients with respect to bryostatin-1’s mechanisms of action. While Kollar and colleagues [137] seem optimistic about the clinical outcomes of these combined therapies including bryostatin-1, the fact remains that bryostatin-1 did not reach a single Phase III clinical trial in oncology about half a century after its discovery by Pettit and his colleagues in 1970. Singh et al. [138] and Irie et al. [139] argue that the limited availability of bryostatin-1 from natural sources and difficulty of synthesis have hampered further large-scale clinical studies, as well as in-depth studies on its mode of action and structural optimization.

### 3.3. Resistance to Pro-Apoptotic Stimuli

In order to metastasize, cancer cells must resist anoikis [140,141,142,143]. Anoikis is an apoptosis-related cell death that is induced in cells that detach from the extracellular matrix (ECM) and/or from neighboring cells. Simpson et al. [144] and Speirs et al. [145] have comprehensively reviewed the multiple biochemical and molecular pathways that enable cancer cells to resist pro-apoptotic stimuli (including anoikis) during their metastatic journey. Drugs that kill cancer cells through pro-apoptotic stimuli are already essential parts of any chemotherapeutic regimen and are, thus, no longer considered novel. However, there is a need to develop innovative novel anticancer drugs that kill cancer cells resistant to pro-apoptotic stimuli (such as metastatic cancer cells) [146]. 

Glioblastoma, a type of brain cancer which does not metastasize, or does so only rarely, is a cancer that displays very high levels of resistance to pro-apoptotic drugs and is therefore associated with dismal prognoses [147]. Interestingly, bryostatin-1 (**1**, Figure 4) was shown to restore sensitivity to pro-apoptotic stimuli in glioblastoma cells [132,133]. 

The cytotoxic effects of tambjamines (**3**, Figure 5), 4-methoxy bispyrrolic alkaloids, isolated from different species of bryozoans (*Bugula longissimi*, *Virididentula dentata*, *Sessibugula translucens*) [79,80,81,82,83], are partly related to their ability to intercalate DNA and to their pro-oxidant activity [148], as has been already demonstrated at the experimental level for tambjamine D (**3d**) [149]. This latter behaves as a poisonous pro-apoptotic compound against normal fibroblasts through increased nitrite/nitrate production and DNA strand breaks associated with genotoxic effects [149]. Therefore, tambjamine D cannot be labeled as a potential/promising anticancer compound. This is true also for other members of the tambjamine family including tambjamines C (**3c**), E (**3e**), F (**3f**), G (**3g**), H (**3h**), I (**3i**), J (**3j**), and K (**3k**) that display similar cytotoxic effects to both normal and cancer cells [82,150]. Interestingly, at least for tambjamine I (**3i**), we do not think that it actually behaves as a classical pro-apoptotic compound. Indeed, tambjamine I (**3i**) was analyzed by the NCI and the mean GI_50_ concentration of **3i** in the NCI 60-cancer-cell-line panel was ~2 µM (https://dtp.cancer.gov/databases_tools/data_search.htm). We ran the NCI-developed COMPARE analysis [151] on tambjamine I (**3i**) and this analysis turned out to be negative with a cut-off value as low as 60% of correlation for **3i** versus all the compounds already included in the Standard Agent database, which includes more than a thousand pro-apoptotic compounds. Tambjamine I, thus, seems to be a promising anticancer compound. 

In addition, some tambjamine synthetic analogs (i.e. **3l**, Figure 5) [152] have shown selective growth inhibitory effects between normal and cancer cells. In particular, compound **3l** displayed anti-invasive properties in human cancer cells at non-cytotoxic concentrations [152].

Perfragilin A (**8a**, Figure 9) and B (**8b**, Figure 9) were isolated from the marine bryozoan *Biflustra perfragilis* (previously *Membranipora perfragilis*). Both display an isoquinoline quinone skeleton as determined by X-ray analyses [153,154,155]. 

Perfragilin B (**8b**) was assayed by the NCI in the 60-cancer-cell line panel and the data obtained are illustrated in Figure 10. These data were obtained from the publicly available Standard Agent database (https://dtp.cancer.gov/dtpstandard/dwintex/index.jsp). How the data in Figure 10 were obtained and presented is explained in Shoemaker [151]. Although perfragilin B (**8b**) displays a mean GI concentration of ~4 µM, it does not behave as a non-selective toxic compound against this panel of 60 cancer cell lines. Indeed, the bar projecting to the left of the mean GI50 concentration in Figure 10 point to those individual cell lines that are less resistant to perfragilin B, while the reverse feature is illustrated by the bar projecting to the right (the most resistant cell lines). Therefore, perfragilin B (**8b**) is a rather selective compound which exerts higher growth inhibitory effects in central nervous system and ovarian (and also some renal) cancers than in leukemia.

We ran a COMPARE analysis for perfragilin B (**8b**) in the NCI Standard Agent database and the highest correlation we obtained was between perfragilin B and caracemide with a weak COMPARE coefficient correlation index (0.6). Therefore, it is not possible to state whether perfragilin B and caracemide share common mechanisms of action. However, the possibility remains that part of the anticancer effects displayed by perfragilin B and caracemide could be common. Caracemide (N-acetyl-N,O-di(methylcarbamoyl)-hydroxylamine; **9**, Figure 9) entered several Phase II clinical trials in oncology during the 1990s, but without evidence of clinical activity, at least in renal [156] and colon [157] cancer patients. However, only a small cohort of cancer patients was treated with caracemide in these two trials and without any selective patient-oriented treatment. Caracemide (**9**) acts as an inhibitor of the ribonucleotide reductase (RNR), an enzyme that catalyzes the reduction of ribonucleotides to their corresponding deoxyribonucleotides and that provides a balanced supply of precursors for DNA synthesis. Hydroxyurea is also an RNR inhibitor that is routinely used in oncology to treat various types of cancer patients [158,159]. The marked but selective growth inhibitory effects displayed by perfragilin B (**8b**) along with a COMPARE-positive association (even if it is weak) with caracemide actually militate in favor of pursuing the characterization of the anticancer effects of perfragilin B. The total synthesis of perfragilins A and B was already successfully performed [160]. The skeleton of perfragilins is reminiscent of that of mimosamycin A (**10**, Figure 9), which is an antibiotic isolated from terrestrial bacteria [161,162] and various marine sponges [163,164]. Perfragilins could thus be part of the microbe-produced chemical defense arsenal of the bryozoan *Biflustra perfragilis*.

## 4. Bryozoan-Derived Metabolites Active in the Brain

### 4.1. Alzheimer’s Disease (AD)

Russo et al. [165] recently wrote that “AD is a multifactorial neurodegenerative disorder and current approved drugs may only ameliorate symptoms in a restricted number of patients for a restricted period of time”, thereby highlighting the dearth of available drugs to treat this devastating disease. Bryostatin-1-mediated activation of PKC isozymes, which was already highlighted to have cognitive restorative and antidepressant effects more than a decade ago [166], was also shown to ameliorate the rate of premature death and improve behavioral outcomes in transgenic mice harboring human AD characteristics [167]. More recently, it was demonstrated experimentally, for the first time, that bryostatin-1 can indeed improve cognition in rodent AD models and “represents a novel, potent and long-acting memory enhancer with future clinical applications in the treatment of AD patients” [168]. Bryostatin-1 was also proposed by Nelson et al. [169] as a potential treatment for AD patients. 

Kororamides A and B (**11a**–**11b**, Figure 11) are tribrominated indolic derivatives isolated from the bryozoan *Amathia tortuosa* [170,171]. A computational study on kororamides as well as on convolutamines (**7**, Figure 8) suggested they could act as tau and dual-specificity kinase inhibitors for Alzheimer’s disease [172].

### 4.2. Post-Stroke Activities

Cerebral ischemia and ischemic stroke, caused by restriction of blood flow to the brain, can cause death as well as life-changing physical and mental damage. Remarkably, experimental work on rats has shown that bryostatin-1, administered within a time window of 24 hours post global cerebral ischemia/hypoxia, can reduce neuronal and synaptic damage and aid recovery of spatial learning and memory abilities through the activation of particular PKC isozymes [173]. Tan et al. [174] further confirmed these experimental observations. They produced acute cerebral ischemia by reversible occlusion of the right middle cerebral artery (MCAO) in rodents and observed that repeated bryostatin-1 administration post-MCAO protected the brain from severe neurological injury post-MCAO. These authors further report that bryostatin-1 treatment improved survival rate, reduced lesion volume, salvaged tissue in infarcted hemisphere by reducing necrosis and peri-infarct astrogliosis, and improved functional outcome after MCAO. It was also shown that the combination of exercise and bryostatin treatment in rats with cerebral cortex infarctions produced by photothrombosis increases local serotonin concentrations in the perilesional area which, in turn, induces greater functional recovery than exercise alone [175].

### 4.3. Antiparkinson Activity

There are very few reports of antiparkinson activity exhibited by bryozoan secondary metabolites. Dashti et al. reported that kororamide B (**11b**, Figure 11) displayed effects on early endosomes when profiled on human olfactory neurosphere-derived cells (hONS) from a Parkinson’s disease patient using a multidisciplinary phenotypic assay [171] 

## 5. Antiviral Bryozoan-Derived Metabolites

### 5.1. Human Immunodeficiency Virus-1 (HIV-1)

The persistence of latent HIV-infected cellular reservoirs represents the major hurdle to virus eradication on patients treated with highly active anti-retroviral treatment (HAART) [176,177,178,179]. Furthermore, successful depletion of such latent reservoirs requires a combination of therapeutic agents that can specifically and efficiently act on cells harboring latent HIV-1 provirus. Perez et al. [176] showed that bryostatin-1, alone or in combination with HDAC inhibitors, could be used in HAART-treated patients to validate the hypothesis that reactivating HIV-1 from latency could purge HIV-1 reservoirs. Diaz et al. [177] confirmed these experimental data and demonstrated that bryostatin-1 reactivates latent viral infection in the NHA (primary astrocytes) and U87 (glioblastoma) cells via activation of protein kinase C (PKC)-alpha and -delta, because the PKC inhibitors rottlerin and GF109203X abrogated the bryostatin-1 effect. These authors also showed that bryostatin-1-induced effects are mediated through the activation of the transcription factor NF-κB. Using in vitro HIV-1 post-integration latency model cell lines of T-lymphoid and myeloid lineages, Darcis et al. [178] demonstrated that PKC agonists (including bryostatin-1) and P-TEFb (positive transcription elongation factor b)-releasing agents alone acted as potent latency-reversing agents (LRAs), and that their combinations led to synergistic activation of HIV-1 expression at the viral mRNA and protein levels. Martinez-Bonet et al. [179] evaluated the effect of various combinations of bryostatin-1 and novel histone deacetylase inhibitors on HIV-reactivation and on cellular phenotype in latently infected lymphocyte (J89GFP) and monocyte/macrophage (THP89GFP) cell lines. These authors observed that combinations between bryostatin-1, panobinostat, and/or romidepsin presented a synergistic profile by inducing virus expression in latently infected HIV cells, rendering these combinations an attractive novel and safe option for future clinical trials.

### 5.2. Chikungunya Virus (CHIKV)

Staveness et al. [180,181] report that Chikungunya virus (CHIKV), which causes devastating arthritic and arthralgic symptoms, has been spreading rapidly, with over one million confirmed or suspected cases in the Americas since late 2013. Currently, there are neither vaccines nor specific treatments for chikungunya, but work using bryostatin-1 analogues has shown that they are potent protective agents against CHIKV-mediated cell death. The same research group showed that these bryostatin-1-mediated effects could be PKC-independent [180].

### 5.3. Polio Virus

Morris and Prinsep [182] reported marked activity (40 µg/well) against polio virus type 1 for amathaspiramide E (**12e**) which belongs to the family of amathaspiramides. Amathaspiramide A–F (**12a**–**12f**, Figure 12) are alkaloids isolated from *Amathia wilsoni* [182] and are believed to be biosynthetically derived from amathamides (**5**, Figure 7) [182]. There are few data published in the literature about their bioactivities, including one report by Morris and Prinsep [182] and the other by Shimokawa et al. [183], whereas several papers have reported on their synthesis [184,185,186,187,188]). 

## 6. Antiparasitic Bryozoan-Derived Metabolites 

### 6.1. Antitrypanosomal Activity

African trypanosomiasis, also known as African sleeping sickness, is a sub-Saharan parasitic disease caused by the protozoan parasite *Trypanosoma brucei*. African trypanosomiasis is mostly transmitted through the bite of infected tsetse flies and, if left untreated, causes debilitating symptoms as the parasite infects the central nervous system (Stage II), including confusion, sensory disturbances, personality changes, disturbance of the sleep cycle, and death. Although there are currently no vaccines, some treatments are available. However, current drugs are either infections stage or subspecies specific (pentamidine and suramin: Stage I of *Trypanosoma brucei gambiense* and *T. b. rhodesiense*, respectively; eflornithine: Stage II of *T. b. gambiense*, only), toxic and resistance-inducing (melarsoprol), or time-consuming to administer (eflornithine) [189]. Thus, novel drugs against trypanosomiasis are urgently needed for the 70 million people at risk. Convolutamines I and J (**7i–7j**, Figure 8), first isolated from the bryozoan *Amathia tortuosa*, were shown to be active against *T. b. brucei* [100]. Amongst the two, convolutamine I (**8i**, Figure 9) was shown to be more active towards *T. b. brucei*, but its high molecular weight of 473 prevents its passage through the blood–brain barrier, thereby making it ineffective against Stage II of the infection [100]. However, subsequent work has produced promising synthetic analogues of convolutamine I, which are small enough to pass through the blood–brain barrier while also retaining their activity against *T. b. brucei* [190].

### 6.2. Nematocidal Activity

Pentaporins A–C (**13a–13c**, Figure 13), which originate from the Mediterranean bryozoan *Pentapora fascialis*, have been shown to have in vitro activity against the nematode *Trichinella spiralis* [191], otherwise known as the pork worm. These compounds possess a dimeric structure (linked by a disulfide bridge) containing different sulfate ester groups. 

Other bryozoan metabolites displaying marked nematocidal activity are convolutamine H (**7h**, Figure 8) and convolutindole A (**14**, Figure 14), this latter isolated from a Tasmanian specimen of *Amathia convoluta* [99]. They are active against the free-living larval stages of the blood-feeding nematode *Haemonchus contortus*, which causes excessive anemia, oedema, and sudden death in infected sheep and other ruminants [99]. In fact, the nematocidal activities displayed by compounds **7h** and**14** were more potent than that of levamisole, a commercially available anthelmintic drug [99]. Narkowicz et al. [99] report that while convolutindole A (**14**), convolutamine H (**7h**), and levamisole (**15**, Figure 14) have some structural similarities, these compounds do not inhibit nematode development by the same mechanisms. Indeed, while levamisole causes characteristic paralysis of nematode larvae, convolutindole A (**14**) and convolutamine H (**7h**) are lethal to the first and second stage larvae of this parasite [99]. Nematocidal, antifungal and antibacterial activities have also been reported for amathamides [192]. 

### 6.3. Anti-Plasmodial Activity

Carroll et al. [193] reported anti-plasmodial activities for wilsoniamine A (**16a**, Figure 15) and B (**16b**, Figure 15) isolated from the Australian bryozoan *Amathia wilsoni*. These compounds possess a hexahydropyrrolo[1,2-c]imidazol-1-one ring system (Figure 15) that had not previously been found in nature. Total syntheses of wilsoniamine A (**16a**) and B (**16b**) were recently proposed by Khan and Ahmad [194]. 

Caulamidines A (**17a**, Figure 16) and B (**17b**, Figure 16) from *Caulibugula intermis* are trihalogenated alkaloids possessing a 2,6-naphthyridine core that showed a similar inhibitory effect against wild-type and drug-resistant strains of *P. falciparum* with IC_50_ values ranging from 8.3 to 12.9 µM [195].

Antiparasitic activity has been also reported for kororamide A (**11a**, Figure 11) by Carroll and co-authors. This compound, isolated from *Amathia tortuosa*, showed a very mild effect against chloroquine-sensitive and -resistant strains of *P. falciparum* [170,196].

## 7. Sourcing of Bryozoan-Derived Compounds of Medicinal Interest

The major obstacle for drug development from bioactive marine natural compounds is the limited supply of these compounds, particularly when molecules that exhibit excellent in vitro activity should be moved to in vivo experiments and clinical trials that need gram levels of pure product [197]. In addition, the amounts of bioactive compounds often suffer from quantitative variability in nature, the difficulties of collecting the source organisms and, sometimes, the stability of the molecules themselves. To overcome these hurdles, several approaches can be actuated, including marine invertebrate aquaculture, synthesis, partial synthesis, and symbiont culture, all of which strongly depend on the biology of the producing organisms and on the true origin of the bioactive compounds. Tian et al. [10] and Wu et al. [11] already comprehensively reviewed this topic. Below, we outline our perspectives on these crucial aspects that are hoped to enable bryozoan metabolites to be used in human medicines.

### 7.1. Harvesting Wild Bryozoans

As components of the sessile fauna, bryozoans inhabit a range of substrates in mostly marine environments, including algae, stones, shells, and other animals (cnidarians, sponges, and other bryozoans), as well as artificial substrates, such as concrete walls, plastic, wood, and metal structures [198]. Bryozoans may be harvested from the aquatic environment by targeted collecting using diving, trawling, or dredging, or as bycatch.

Many bryozoan species, including *Bugula neritina*, *Virididentula dentata*, *Watersipora subtorquata*, *Cryptosula pallasiana*, *Amathia convoluta*, and *Amathia verticillata*, are well-known fouling species and are widespread around tropical and subtropical waters, mainly in harbor areas [43,75,199,200]. Colonies of these species may have high biomass and be easily collected by hand or be scraped off the substrate using a spatula. Unfortunately, most of the compounds are found in only low levels in the bryozoan tissue. In one example of a large-scale collection of bryostatin-1 (**1**, Figure 4), approximately 28,000 pounds wet weight of *B. neritina* were collected and extracted, yielding 18 g of bryostatin-1 [201]. Obviously, this is not ecologically sustainable. However, since *B. neritina* larvae were shown to have 10× higher levels of bryostatins compared to the adults [58], the production of larvae and adults in a laboratory with maintenance of adult brood stocks may help in studies of bioactive substances [202].

Lastly, the yields of a given bryostatin can vary markedly according to geographical location [64,203], which is likely the result from distinct *Bugula–*symbiont associations [74]. This feature can in turn explain, at least partly, why not all bryostatins are found in all *Bugula neritina* specimens [204]. For example, while bryostatin-1, -2, and -3 are mainly found in *Bugula neritina* specimens from the Eastern Pacific Ocean, bryostatin-4 is found in *Bugula neritina* specimens from the Western Pacific Ocean, the Gulf of California, and the Gulf of Mexico [205]. Ueno et al. [206] did not find bryostatin-1 in Japanese specimens of *Bugula neritina* collected in the Gulf of Imazu (Fukuoka, Japan). Thus, harvesting from the wild adds a certain quantitative and qualitative unpredictability to the yield.

### 7.2. Culturing Bryozoans

As many bioactive natural products found in marine invertebrates are produced by microbial associates [207,208], there are several approaches that could result in more sustainable levels of the bioactive compounds, one of which is culturing of the producing organism followed by induction of compound production. Unfortunately, this can be fairly difficult for true symbiotic microbes, which have evolved to rely on some aspects of their hosts’ metabolism. Advances in sequencing technologies have enabled the sequencing of the genomes of microbial producers, which can allow researchers to predict what compounds may facilitate culturing the microbe. The genome of the bryostatin-producing *B. neritina* symbiont, “*Candidatus* Endobugula sertula” was sequenced [209]; unfortunately, the authors were not able to speculate on what compounds may be missing from the culture media.

### 7.3. Heterologous Production

If the natural compound producing microbe is recalcitrant to cultivation, another approach is to clone and heterologously produce the compound in a different host. This has been accomplished for some bacterial polyketide metabolites such as epothilone [210,211] and erythromycin [212], but it remains a challenging prospect. First, some of the gene clusters that prescribe biosynthesis are very large (the *bry* gene cluster in “*Candidatus* Endobugula sertula” that encodes the biosynthetic machinery to make a common precursor of bryostatins is ~77 kb long [213]), and cloning and expressing DNA fragments that long can be difficult. Secondly, DNA from symbiotic bacteria are typically A/T rich [214], which can result in DNA instability or expression problems in the host. Finally, the host must have an abundance of the correct precursors. There are ongoing efforts to engineer strains that can produce metabolites from other organisms heterologously [215,216,217,218], but the challenges are considerable, and can differ depending on the metabolite.

### 7.4. Compound Extraction

Bryozoan metabolites have so far been isolated through the application of traditional protocols, based mainly on the use of organic solvents. Recently, the search for new approaches, mainly aiming at improving the yield of bioactive compounds, has led to the development of different extraction systems that, compared to the traditional ones, can reduce not only time but also hazards associated with the exposure to the solvents. These methods include microwave-assisted extraction (MAE), ultrasound-assisted extraction (UAE), the application of supercritical fluids (SFE), pressurized solvent extraction (PSE), pulsed-electric field-assisted extraction, the use of enzymes, or the application of ionic solvents. A comparison of the yield of high-value compounds, after application of these different extraction protocols [219], has been done for many invertebrates such as macroalgae, microalgae, crustaceans, and sponges. Concerning bryozoans, that include both uncalcified and calcareous species, no information on the employment of these new technologies has been found in the scientific reports to date. We think that with the help of new practices that assist with the disruption of cell membranes, the efficiency of extraction can be greatly augmented while the time and the noxious exposure are reduced, even if the costs of these advanced technologies are still fairly elevated. In addition, the use of so-called “smart solvents”, which include polarity- and hydrophilicity-switchable solvents and ionic liquids, should also be beneficial in productive bryozoan metabolite extraction [219].

## 8. Synthesis

### 8.1. Chemical Synthesis

Synthetic organic chemistry has often been touted as the solution to the supply problem of sufficient amounts of bioactive compounds from marine invertebrates for drug development. However, the structural and stereochemical complexity of bioactive compounds remains a significant challenge in this regard [220,221]. Bryostatin-1 synthesis, for example, requires 60 steps [221]. There is thus some irony in the fact that the same complexity which results in a plethora of bioactivities, often with different modes of action, also results in synthetic pathways consisting of >40 chemical steps which are generally not considered short enough for realistic pharmaceutical exploitation. 

Numerous groups are searching for reasonable solutions to these significant challenges. Kornienko and co-workers recently published a review with an in-depth analysis of the supply problem affecting marine-based bioactive natural products with anticancer activity [222]. Possible solutions include the following broad research themes concerning compounds derived from the bryozoans:
a)Further research into the development of shorter synthetic routes that produce bioactive compounds for downstream exploitation by the pharmaceutical industry.b)Increased investigations into simplified analogues of bryozoan origin. The design and synthesis of structurally and stereochemically simplified bioactive compounds inspired by natural products will in itself result in shorter synthetic pathways that should be amenable to scale-up. These simplified analogues could be initially accessed through synthetic derivatizations and manipulations, or through modification of biosynthetic pathways.c)Increased development of processes aiming at the production of bioactive compounds or advanced biosynthetic intermediates. Multidisciplinary collaborations involving the fields of genetics [223,224], microbiology [225,226], and bioengineering [227] should allow for the identification of the microbial producers of important compounds, the genetic “up-scaling” required for production of the compounds and the expertise needed for large scale production and subsequent isolation of the desired bioactive compounds. These strategies are likely necessary to replace the large-scale cultivation/collection of marine organisms, utilized for some marine-derived compounds which, to date, have often resulted in enough compound for clinical trials (but unfortunately not enough for clinical application) [228]. It should also be noted that some of these biotechnological approaches could be aimed at the production of advanced intermediates which could then be utilized further as semisynthetic strategies to afford the bioactive natural products or their derivatives/analogues. For reviews on this theme regarding accessing the bryostatins, see [228,229]. Wu et al. (2020) recently published a comprehensive review entitled “Unlocking the Drug Potential of the Bryostatin Family: Recent Advances in Product Synthesis and Biomedical Applications”. It could be argued that as more compounds of bryozoan origin are being identified as having valuable medicinal application (essentially on a yearly basis [9,230]), the pressure of combining advanced biotechnology and synthetic organic chemistry to provide significant amounts (gram to kilogram scale for clinical trials and even larger amounts for clinical application) will surely increase. It is thus up to the combined fields of natural products isolation and characterization, biochemical and biological evaluations, microbiology, biotechnology, and synthetic chemistry to come up with solutions for this important challenge.

### 8.2. Partial Synthesis

Various authors already reviewed a large number of publications relating to the partial syntheses of bryostatins along with their bioactivities [220,231], with an emphasis on their effects on PKC signaling [139,232,233]. Pettit and colleagues performed a synthetic conversion of bryostatin-2 to bryostatin-1 [234] and, also, modified the chemical structure of bryostatin-2 in order to perform structure–activity relationship (SAR) analyses [235].

### 8.3. Total Synthesis

None of total syntheses of bryostatins meet the minimal costs of goods (COGs) requested by pharmaceutical groups to succeed with the potential marketing of a given bryostatin, even if a given bryostatin demonstrated clear benefits for cancer patients in Phase II and Phase III clinical trials. Biotechnology-related approaches should be investigated in depth in solving this problem, as detailed in Section 7.2. The synthesis of selective anti-PKC bryostatin analogues would also be a very elegant manner in which to solve this problem.

### 8.4. Biosynthesis

Because of the high pharmaceutical potential of bryostatins and their low natural concentrations, the discovery that an associated microbial symbiont produces bryostatin led to much effort being put into the discovery and characterization of the bryostatin biosynthetic gene cluster by Sherman and colleagues [236] with the goal of heterologously producing bryostatins in a microbial host. *BryA*, the ORF containing the putative loading module and first polyketide extension modules, was cloned and sequenced from California populations of *B. neritina* [236]. The bacterial symbiont that produces bryostatins has thus far been recalcitrant to cultivation, which prevents gene knockouts that could confirm the PKS gene cluster responsible for bryostatin biosynthesis, which led to later efforts focused on overexpressing portions of the gene cluster heterologously and demonstrating their activity in vitro. The discrete acyltransferase *bryP* was overexpressed in *E. coli* and shown to load the extender unit, malonyl-CoA, onto ACPs excised from the bryostatin PKS cluster, as well as a complete *bry* PKS module [237]. In addition, *bryR*, the HMG-CoA synthase, was also overexpressed in *E. coli* and was shown to be capable of producing a β-branched intermediate that could result in the pendant methyl ester groups at C13 and C21 of the bryostatin macrolactone [237]. While not explicitly demonstrating that the *bry* cluster prescribes biosynthesis of bryostatin, these studies support this hypothesis. 

## 9. Conclusions

Bryozoans are a source of many pharmacologically important bioactive compounds, many of which may be ecologically relevant. These compounds have been shown to affect predator–prey interactions, competition, and host–symbiont relationships. Understanding how natural products affect these significant members of benthic communities will shed more light onto the diversity and potential of bryozoan-associated secondary compounds. The high potential of bryozoan metabolites as anticancer agents has been recently reviewed by Wu et al. (2020), Tian et al. (2018), and Figuerola and Avila (2019). In the current review, additional information about bryozoan metabolites with potential for overcoming cancer chemoresistance have been discussed. This review has also highlighted the potential of bryozoan metabolites as treatments for brain diseases and viral and parasitic infections. The review concludes with thoughts on overcoming the challenges of isolating compounds from bryozoans due to the low sample masses provided by these invertebrate organisms. Amongst potential solutions are novel approaches to culturing and harvesting of bryozoans, as well as utilization of advanced extraction methods. In addition, the value of organic chemistry in the development of novel bryozoan-inspired analogues and derivatives is highlighted, specifically within the concepts of partial, total, and biosyntheses. The aim of this review is to stimulate research by interdisciplinary teams of medics, biochemists, chemists, organismal biologists, and taxonomists to better utilize the vast array of known and unknown bioactive compounds found in bryozoans for the development and marketing of novel drugs.

## Figures and Tables

**Figure 1 marinedrugs-18-00200-f001:**
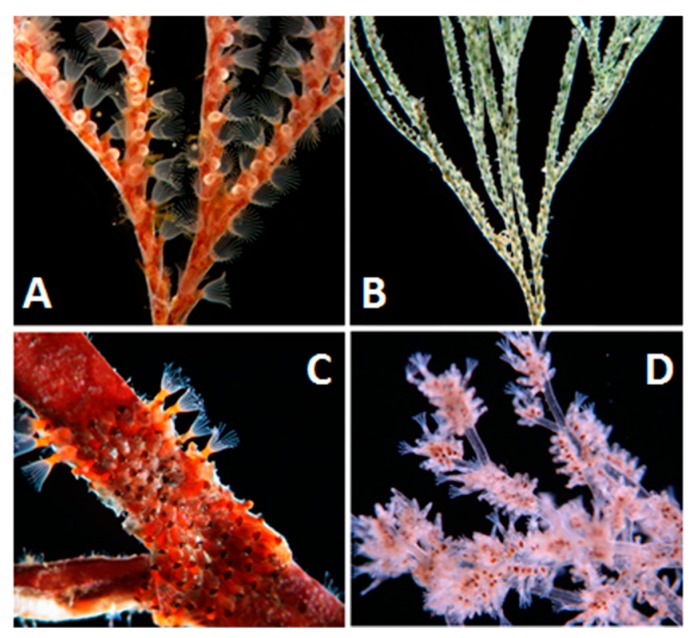
Illustration of different bryozoan colony forms. Arborescent: **A**. *Bugula neritina*, **B**. *Viridentula dentata*. Encrusting: **C**. *Watersipora subtorquata*. Arborescent: **D**. *Amathia verticillata*.

**Figure 2 marinedrugs-18-00200-f002:**
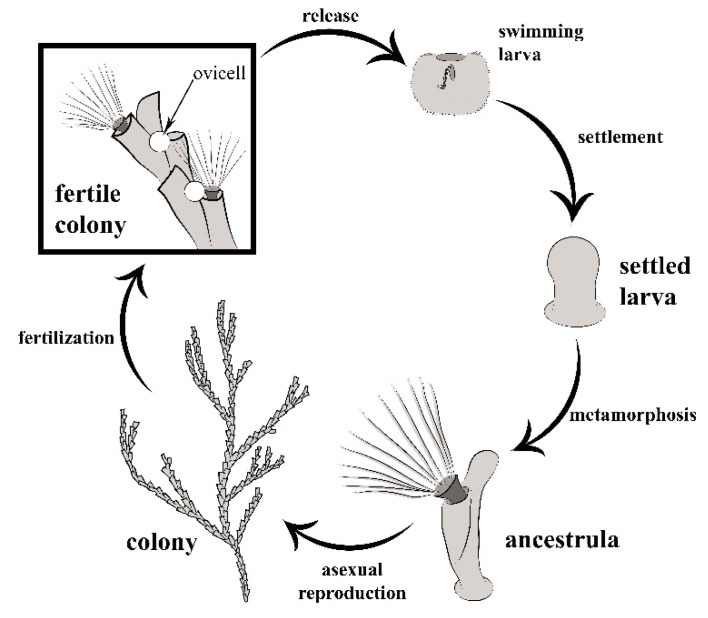
Schematic illustration of the life cycle of *Bugula neritina*. After fertilization, the embryos are brooded in ovicells until they are released as non-feeding larvae. The larva settles and metamorphoses into the founder zooid of the colony (ancestrula) which gives rise to the rest of the colony by asexual budding.

**Figure 3 marinedrugs-18-00200-f003:**
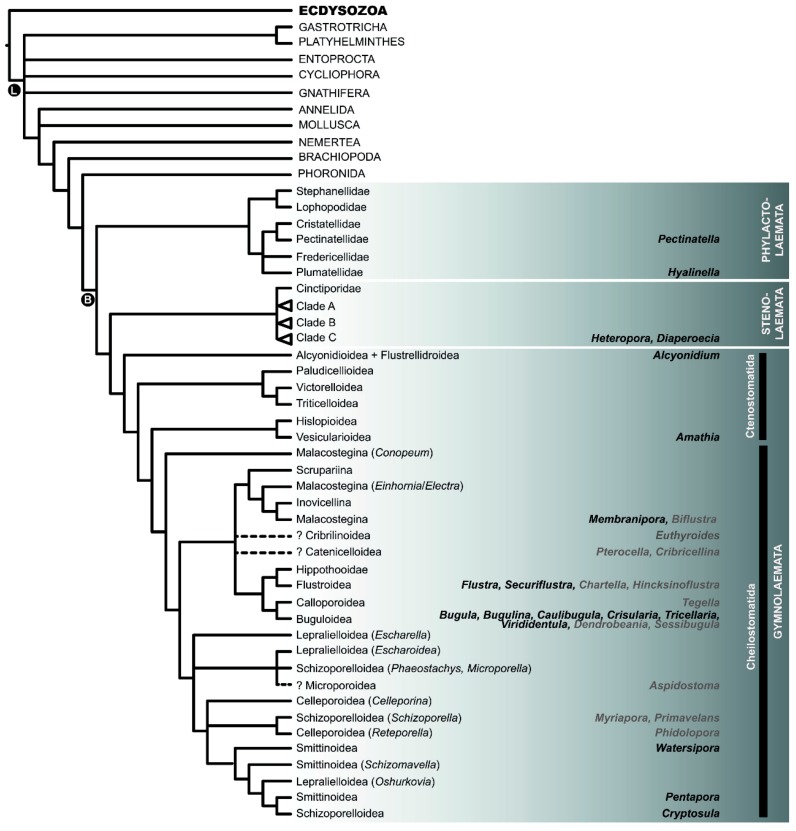
Cartoon summarizing recent results regarding the position of the Bryozoa on the tree of life based on Laumer et al. [25] and Nesnidal et al. [26], and the interrelationships within the Bryozoa based on Knight et al. [39], Laumer et al. [25], Nesnidal et al. [26], and Waeschenbach et al. [24,27]. Bryozoan terminals are given as higher-level classification (Phylactolaemata: families; Stenolaemata: families or Clades A–C (for clade details, see Waeschenbach et al. [24]; Ctenostomatida: superfamilies; Cheilostomatida: suborders (Malacostegina, Inovicellina, Scrupariina) or superfamilies). In cases where higher taxonomic groups were not monophyletic, genera representing those lineages are given. Uncertainty in the placement of lineages is indicated by dotted lines and question marks. Genera that have been shown to harbor bioactive compounds are given on the right-hand side in bold. Genera which were already included in phylogenetic analyses are given in black. Genera in grey have not been put into a phylogenetic framework, yet. They are placed in approximate positions based on their higher-level classification. L = Lophotrochozoa; B = Bryozoa.

**Figure 4 marinedrugs-18-00200-f004:**
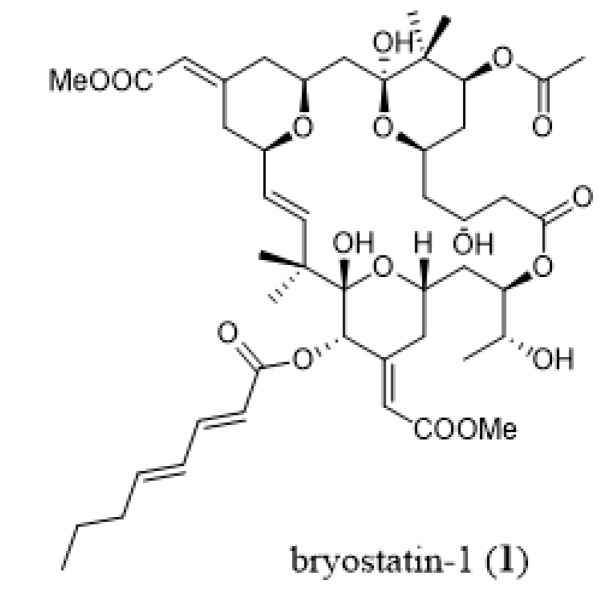
Chemical structure of bryostatin-1 (**1**).

**Figure 5 marinedrugs-18-00200-f005:**
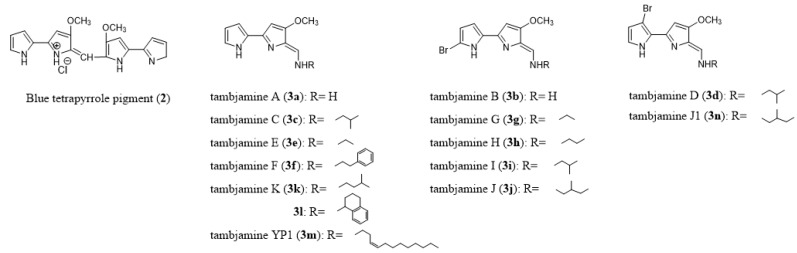
Chemical structures of blue tetrapyrrole pigment (**2**) and natural and synthetic analogues **3a**–**3n** of tambjamines.

**Figure 6 marinedrugs-18-00200-f006:**
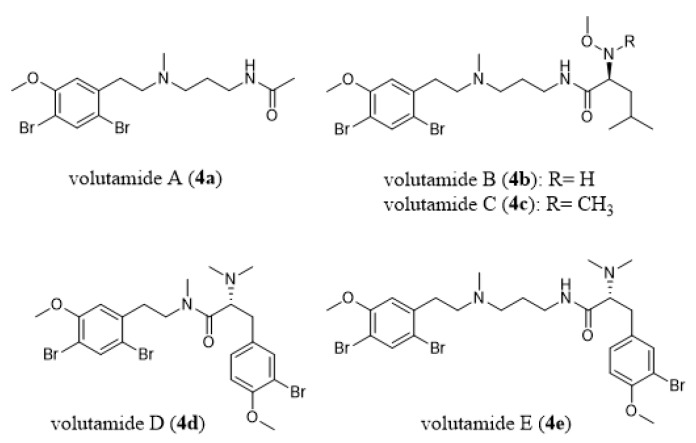
Chemical structures of volutamides A–E (**4a**–**4e**).

**Figure 7 marinedrugs-18-00200-f007:**
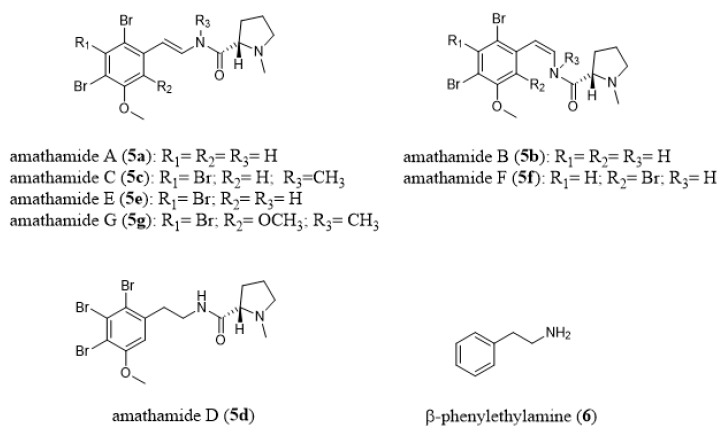
Chemical structures of amathamides A–G (**5a**–**5g**) and β-phenylethylamine (**6**).

**Figure 8 marinedrugs-18-00200-f008:**
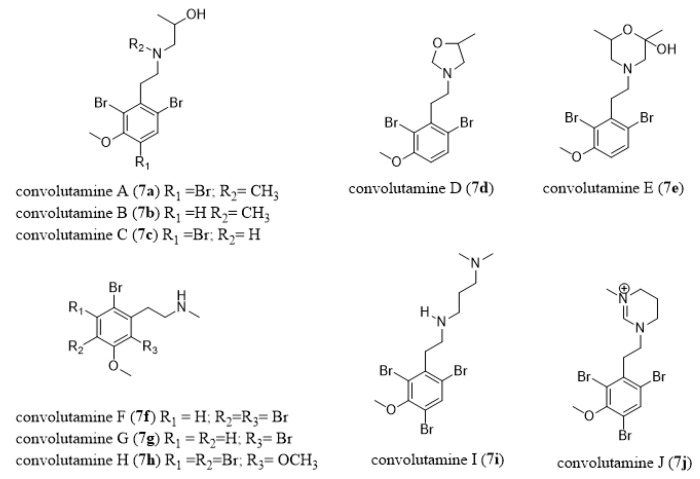
Chemical structures of convolutamines A–J (**7a**–**7j**).

**Figure 9 marinedrugs-18-00200-f009:**
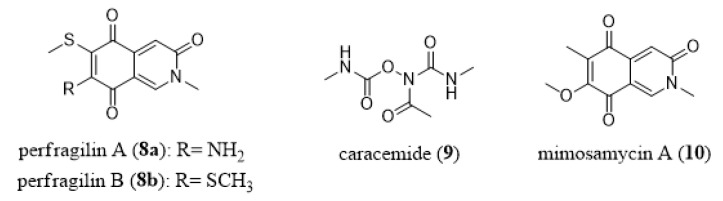
Chemical structures of perfragilin A (**8a**) and B (**8b**), caracemide (**9**), and mimosamycin A (**10**).

**Figure 10 marinedrugs-18-00200-f010:**
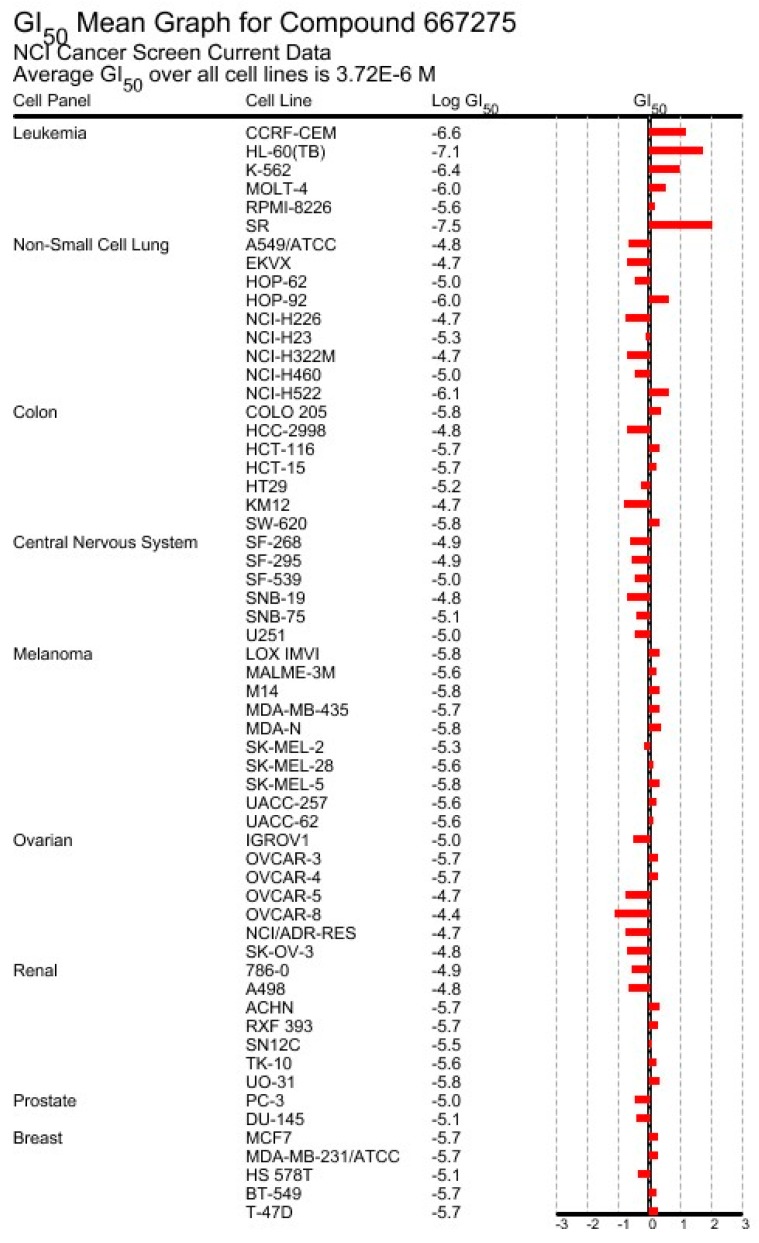
Growth inhibition profile of perfragilin B (**8b**; NSC667275) in the NCI 60-cancer-cell-line panel. Each dose–response curve (5 doses tested from 0.01 to 100 µM) obtained for perfragilin B (**8b**) on a given cell line enables the GI_50_ concentration to be calculated by linear interpolation. The mean GI_50_ concentrations calculated on the 60 cancer cell lines is represented by the vertical black bar, which indicates that perfragilin B (**8b**) displays a mean GI_50_ concentration of ~4 µM.

**Figure 11 marinedrugs-18-00200-f011:**
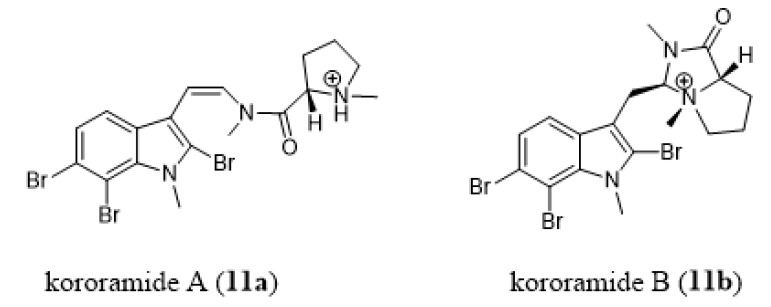
Chemical structures of kororamide A (**11a**) and B (**11b**).

**Figure 12 marinedrugs-18-00200-f012:**
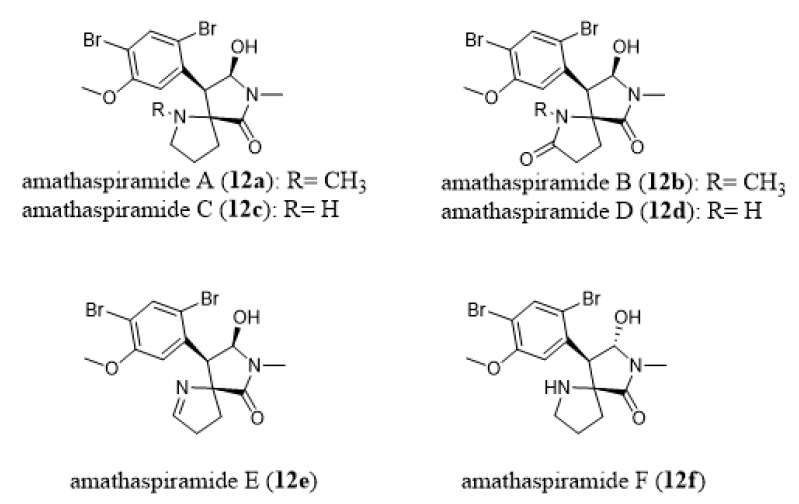
Chemical structures of amathaspiramides A–F (**12a**–**12f**).

**Figure 13 marinedrugs-18-00200-f013:**
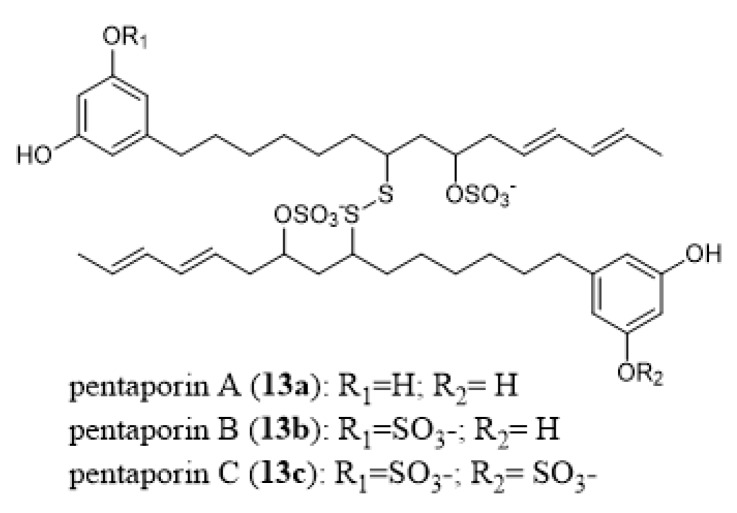
Chemical structures of pantaporins A–C (**13a**–**13c**).

**Figure 14 marinedrugs-18-00200-f014:**
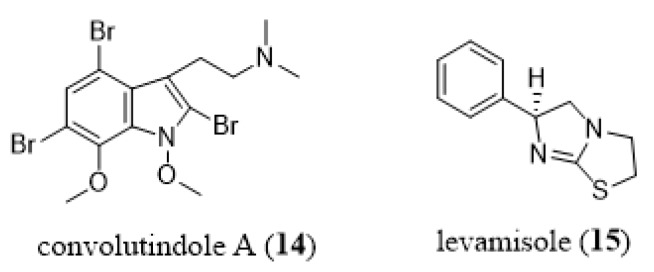
Chemical structures of convolutindole A (**14**) and levamisole (**15**).

**Figure 15 marinedrugs-18-00200-f015:**
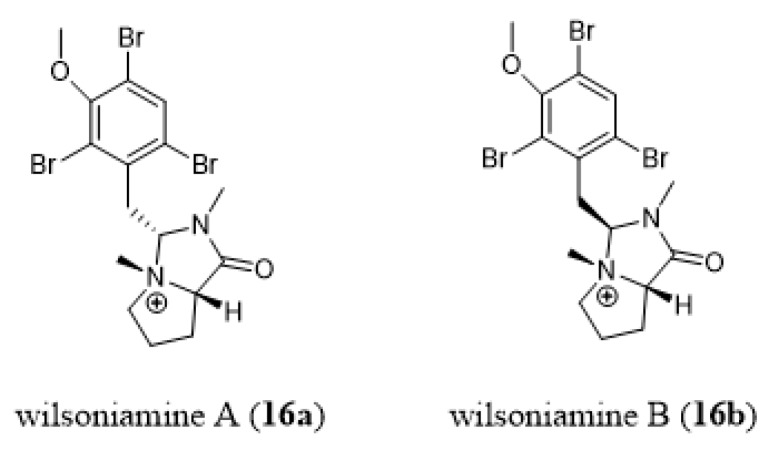
Chemical structures of wilsioniamine A (**16a**) and B (**16b**).

**Figure 16 marinedrugs-18-00200-f016:**
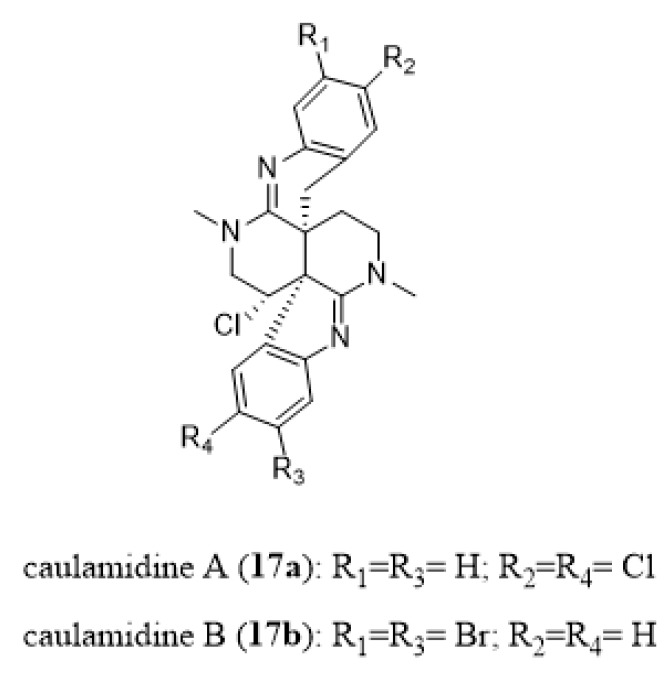
Chemical structures of caulamidine A (**17a**) and B (**17b**).

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
