# Peer review of "The Phylum Bryozoa: From Biology to Biomedical Potential"

_marinedrugs, 2020, doi:10.3390/md18040200_

Round 1

Author Response

Dear Reviewer,

Here attached is the new version of our review (MS #731209) entitled “The phylum Bryozoa: from biology to biomedical potential, which was previously entitled “Bioactive compounds from bryozoans against cancer, viral and brain and parasitic diseases”, by Maria Letizia Ciavatta, Florence Lefranc, Leandro M. Vieira, Robert Kiss, Marianna Carbone, Willem van Otterlo, Nicole B. Lopanik and Andrea Waeschenbach.

Our revised version has been modified according to each Reviewer’s comment as detailed below.

Reviewer 1:

Bioactive compounds from bryozoans against cancer, viral, and brain and parasitic diseases 

Firstly, I want to apologise to the authors for the fact I neglected to notice that when I suggested adding “viral” to the title in my initial review, that I had duplicated “and” (see above correction in title). This is now a very comprehensive, mainly reasonably well written manuscript, but there are a few significant problems I have encountered in reading: Figure 3 is missing (on Page 5 between lines 199 and 200) although the caption is present.

Our response: the title has been modified according to the request of Reviewer 4. We checked that Fig. 3 is provided with the text.

Also, there are references to Section 12.1.1 and 12.1.2 in the text, but no such Section headings appear.

Our response: the numbering of the sections has been carefully checked. We reduced by more than 50% the length of our review and several sections have been deleted in order to satisfactorily respond to the comments of the remaining Reviewers.

I am unconvinced that the amount of detail included in Section 5.2, in particular, (regarding PKC isoforms) in an attempt to justify patient selection criteria in (particularly bryostatin-1) clinical trials is appropriate in this review, given that a number of the examples given for activity data regarding drug interactions with specific isoforms are synthetic drugs rather than of bryozoan or even marine origin. I’ll illustrate with just one of several examples: lines 892-896: How are the facts that Aurothiomalate (a synthetic drug), entered Phase I clinical trials, its dose and mode of dosing of relevance to the topic of this review? This review, I believe, has currently overdone the obvious emphasis on bryostatins, and stating that “we aim to rehabilitate bryostatin-1 as an innovative targeted agent for specific subgroups of patients suffering certain types of cancers” is inappropriate use of the review in my opinion, and it should be removed. There is also an area of repetition in different sections that needs to be removed (please see the following list).

Our response: we have deleted from the revised version of our review the section relating to PKCs. We did the same for bryostatin-1.

Specific issues and suggestions (many are minor):

Page 2, line 59: delete the full stop between “species” and “[9]” and replace with a space.

Page 4, lines 153-154: One of the "However"s that start the two successive sentences here should be removed - I suggest removing the first one, and combining the two sentences into a single sentence - i.e.: "Construction of...in progress, however, for the time being, the taxon...often weak."

Page 5, line 161: “monophyletic”

Page 7, line 285 (Table 1): The column with the “Family” heading needs to be widened so the “e” at the end of Aspidostomatidae and Membraniporidae fits on a single line.

Page 10, line 364: The numbering of Fig. 18 is out of numerical sequence here: the last Figure discussed was Fig. 3. Figure 18 should be renumbered as Figure 4 and moved from page 41 to somewhere here (around page 10). All mentions of Fig. 18 on this page (lines 364, 368 and 374) and on page 40 (lines 1475, 1477, 1486 two places, 1491 and 1495) should be to Fig. 4. Figure numbers and referals to those figures in the text from the current figures 4 to 17 will then all need to be incremented by one.

Our response: we actually thank Reviewer 1 for the impressive work done in analyzing our review. Most of the text mentioned by Reviewer 1 has been deleted from the revised version of our review. However, as indicated below, we made our best efforts in correcting the “surviving” text as suggested by Reviewer 1.

Page 11, line 443: Re: reference 109. I cannot work out why you have specifically chosen the 2018 annual review of Marine Natural Products as one of your supporting reference here, when these reviews have been published annually in Natural Product Reports since the 1970’s and the 2019 update is already in your reference list as reference [9]. You should add reference [9] to your references here, leave reference [109] as well (because, if you remove it, all subsequent reference numbers will need to be decreased by one), but make it a general reference for the series by adding "See also previous reviews in this series" after reference [109] on page 57, as updates in this review series normally start with the reference for the previous one.

Our response: we made our best efforts to modify our references as suggested by Reviewer 1 (see ref. 9).

Page 13, lines 479, 489, 505; Page 16, line 619 and 630; Page 18, line 728; Page 19, line 741 (footnote), 746, 749, 753, 766, 771 two places, 777; Page 24, lines 940 and 956: Fig. 4 now becomes Fig. 5.

Page 16, line 623: (To avoid “and…and…and…”) “…properties. It has even…”

line 626: Figure 4 now becomes Figure 5.

Page 17, line 640: Spacing issue caused by right and left justification.

Page 17, lines 641: I think the in vivo data discussion (lines 670-674) should follow the in vitro discussion directly after [190]. (line 641); then the PKC discussion that follows would lead into lines 674-679 (Section 3.1.1…stem cells [123].) I think the statement “We will detail in Section 5.3 the interplay between bryostatins and PKCs” would come after that.

lines 646-669: Delete – this was all stated verbatim on Page 3.

Page 18, line 721: "has not reached" would be better.

lines 724-725: “…as well as in-depth…”

Pages 18-19, Table 3: The last reference mentioned in the text was [235]. There are problems with either the location of Table 3 here or the reference numbering order when [248], [459], [490], [519], [554], [555], [556] all appear here in Table 3 long before most are mentioned in the text.

Page 19, line 243: “about10” Insert space.

Pages 20-21, Table 4: Similar to the case with Table 3, the last chronological reference mentioned in the text was [259], but Table 4 here also contains lots of reference numbers > [500], many pages before they are mentioned in the text. These references probably should be renumbered in chronological order as they appear in the Table.

Page 22, lines 828-831: The sentence: “A few years later... proved to be negative” needs simplification – it is too hard to follow the sequence of ideas: what proved to be negative?

lines 856-857: “…enabling to quantify hyperactivation versus hypoactivation of PKC isoforms…”??

line 866: “…the anti-PKCα agent, aprinocarsen, …” would be better than “the aprinocarsen anti-PKCα agent”

Page 23, line 888: “…cells that included…”??

lines 904-905: This 2-line statement adds very little here; relocated it to page 50, around line 1904 with the discussion of Bryostatin synthetic analogues to avoid having to change the Fig. 21 number here. The statement that “Irie et al. [235] reviewed the bryostatin analogues that target PKCδ” can be moved from Page 24, line 930 to here, or remain in its current place (without reference to synthetic analogues).

lines 920-922: This doesn’t need to be said again: the sentiments regarding bryostatin-1 clinical trial selection criteria and PKC activities have already been hammered home: see page 18, lines 709-710 and lines 721-722; page 22, lines 837-841.

Page 24, lines 951, 968, 981 and 987: Fig. 5 is now Fig.6

Page 25, line 979 (Figure caption): Figure 5 is now Figure 6

line 987: I suggest – “symmetric disulfide dimers” (to indicate the mode of dimerization).

lines 991 and 992: Spacing issue caused by right and left justification.

lines 991, 992, 993, 1001; page 26, line 1015; page 27, line 1031; page 44, line 1627; page 47,

line 1739: Fig.6 is now Fig 7

Page 26, line 1010: Figure 6 is now Figure 7

line 1030: Murrayanolides (There is only one known)

Page 27, line 1056; page 28, line 1069 two places; line 1075; page 28, line1099: Fig. 7 is now Fig. 8

Page 27, line 1062: Figure 7 is now Figure 8

line 1064 and structure in the Figure: Is there a reason why 10n has been omitted here and the 14th structure is 10o?

Our response: once more, we actually thank Reviewer 1 for the impressive work done in analyzing our review. And as also already indicated above, we made our best efforts in correcting the text as suggested by Reviewer 1, but also by Reviewers 2, 3 and 4, while most of the text and figures referred above by Reviewer 1 has been deleted from the revised version of our review.

Page 28, line 1078: “…towards the nematode Caenorhabditis elegans.” (Most readers won’t know what type of organism C. elegans is).

Our response: we have added nematode in the revised version each time this precision would be of interest for the readers.

line 1092: “We ran…”

line 1099: “…as it is the case…”

lines 1106, 1107, 1109; page 29, lines 1120, 1124, 1137 two places; page 45, line 1643:

Fig. 8 is now Fig. 9

Page 29, line 1117: Figure 8 is now Figure 9

Figure: In the structure for Caulibugulones E and F, NR2 should be NR

line 1127: Spacing issue caused by right and left justification.

line 1140: IC50

line 1143 two places, page 30, line 1148: Fig. 9 is now Fig. 10

Page 30, line 1146: Figure 9 is now Figure 10

line 1151: Fig. 10 here should remain unchanged (as the structure of aspidazide A is in your current Fig. 9, not 10).

line 1158: Fig. 10 is now Fig. 11

Page 31, line 1183: Figure 10 is now Figure 11

Our response: most of the text and figures referred to above by Reviewer 1 have been deleted from the revised version of our review. In contrast, we have corrected the “surviving” text as requested by Reviewer 1.

line 1190: “…cancers such as metastatic cancers that resist pro-apoptotic stimuli such as metastatic cancers.”

Our response: this text has been deleted from the revised version.

line 1198: “…two-order of magnitude GI50 variation in the NCI 60-cancer-cell-line panel”

Our response: this text has been deleted from the revised version.

Page 32, lines 1207, 1212, 1213, 1214 two places, 1220, 1224-1225 three places: Fig. 11 is now Fig. 12

lines 1220-1221: “…not Trp-P-2 (19e), a γ-amino-β-carboline (Fig. 12)…” Note also: (19e) is actually in your Fig. 11, not Fig. 12, so “(Fig. 12)” here should not be altered.

line 1228: Figure 11 is now Figure 12

lines 1230-1231: “…(20a), and 1-ethyl-4-methylsulfone-β-carboline (20b)…”

lines 1233, 134, 1236-1237 three places, 1239 three places 1240; page 33, line 1254: Fig. 12 is now Fig. 13

Page 33, line 1252: “eusynstyelamide-induced”

line 1254: “A- C” Delete the space.

line 1264: Figure 12 is now Figure 13

line 1268: “…we cannot draw…” would be much better than “…it cannot enable us to draw…”

line 1271 two places; page 34, lines 1279, 1282, page 35, 1309 two places, 1310: Fig. 13 is now Fig. 14

Page 34, line 1275: “…solvent-dependent equilibria…”

line 1280: “…showing none significant activity

line 1289: Figure 13 is now Figure 14

Page 35, lines 1314, 1320 two places, 1330, 1333; page 36, lines 1346, 1349; page 45, line 1648: Fig. 14 is now

Fig. 15

line 1323: “…flustramines E (29e) [394], B (29b) [395,396], and A (29a) [397], and also…”

Page 36, line 1338: Figure 14 is now Figure 15

line 1342: “…that enable them to defend themselves…”

lines 1343 to 1344: “Biofouling is the settlement and growth of microorganisms such as bacteria on other organisms [392]”

Our response: most of the text and figures referred to above by Reviewer 1 has been deleted from the revised version of our review.

The wording here needs modification. This “definition” only covers one (small) aspect of Biofouling: In general, biofouling or biological fouling is the undesirable accumulation of micro-organisms, plants, algae, or animals on submerged structures, such as piers, pipes, or ship hulls.

Moreover, when biofouling occurs on the surface of another living marine organism it is commonly referred to as epibiosis rather than biofouling.

Our response: this section does not exist anymore in the revised version.

lines 1352-1353: “…provided evidence of a distribution gradient of these metabolites…”

line 1356; page 37, lines 1363 two places; page 38, line 1398; page 39, line 1412, 1416: Fig. 15 is now Fig. 16

line 1359: “…isolated already three decades ago…”

Page 37, line 1363: “Perfragilins A (32a…”

line 1369: Figure 15 is now Figure 16

line 1372, 1374, 1378, 1382: Fig. 16 becomes Fig. 17

Page 38, line 1386: Figure 16 is now Figure 17

line 1396: perfragilin

Page 39, line 1408: “…actually mitigate in favour…”

lines 1425 two places, 1426, 1428, 1429, 1430, 1431, 1443; page 40, lines 1456, 1469; page 47, line

1732, 1742 two places: Fig. 17 becomes Fig. 18

Page 40, line 1453: Figure 17 is now Figure 18.

lines 1475, 1477; page 41, lines 1486 two places, 1491, 1495, 1502: Fig. 18 is now Fig. 4

Page 41, line 1493: amathamide G

lines 1493-1494: Fig. 19 (which should have been Fig. 18) is now Fig. 4

line 1497: amathamides D and F

Page 42, line 1554-1555: “…that has not previously been found in nature.” Is this important to include? In the preceding paragraph, you discussed Kororamine B that possessed a hexahydropyrrolo[1,2-c]imidazolone ring system: it is factual that the Wilsoniamine structures were published prior to that of Kororamine B, but still seems odd in a review to make a point of saying “these are the first” immediately after presenting another one! If it is important, you might consider presenting the Kororamine and Wilsoniamine paragraphs in the reverse order, but that would require alterations to Section, structure and reference numbers.

Our response: this section does not exist anymore in the revised version.

Page 45, line 1639: “…access to to small…”

line 1640: “…that the the…”

line 1651: aselective Insert space

line 1642: “therefore merits” Remove large space (or is this a right/left justification issue?)

line 1670: Remove the space before the full stop

Page 47, line 1734: “…prevents it from passing…” would be better than “prohibits it to pass”

Page 49, line 1833: “…with the disruption of cell…”

line 1862: “Section 12.1.1” There is no such section in this review

line 1894: “Section 12.1.2” There is no such section in this review

Page 52: line 1971: “In the author’s opinions…” Which author? Or do you mean “In the authors’opinion”? (…given that only one opinion is given here)

Our response: most of the text and figures referred to above by Reviewer 1 has been deleted from the revised version of our review.

Page 61, line 2468: Is there a reason for the unusual format for ref. 190: “Nat. Rev. Cancer 2006;6:813-823.” rather than “Nat. Rev. Cancer 2006, 6, 813-823.”?

Our response: the reference has been corrected.

Hoping that the re-revised version of our review will be acceptable for you, I remain,

Sincerely Yours,

Florence Lefranc, MD, PhD

Reviewer 2 Report

None of the issues previously identified in the original review by this reviewer have been addressed.  Therefore the assessment of the paper has not changed.

Author Response

Dear Reviewer,

Here attached is the new version of our review (MS #731209) entitled “The phylum Bryozoa: from biology to biomedical potential, which was previously entitled “Bioactive compounds from bryozoans against cancer, viral and brain and parasitic diseases”, by Maria Letizia Ciavatta, Florence Lefranc, Leandro M. Vieira, Robert Kiss, Marianna Carbone, Willem van Otterlo, Nicole B. Lopanik and Andrea Waeschenbach.

Our revised version has been modified according to each Reviewer’s comment as detailed below.

Reviewer 2:

None of the issues previously identified in the original review by this reviewer have been addressed. Therefore, the assessment of the paper has not changed.

Our response: we hope that this time we satisfactorily responded to Reviewer 2. Indeed, we deleted 80% of the sections relating to cancer in the revised version of our review. We thus reduced by more than 50% the size of or review as requested by Reviewer 2. For the cancer topics, we now refer to the reviews published by Tian et al. (2018), Wu et al. (2019) and Figuerola and Avila (2020). We kept only a reduced version on cancer including topics that were not covered by the three above mentioned reviews. We removed from the revised version all the sections relating to PKCs, as also requested by Reviewer 1.

Hoping that the re-revised version of our review will be acceptable for you, I remain,

Sincerely Yours,

Florence Lefranc, MD, PhD

Reviewer 3 Report

The present review offers a very comprehensive report of all chemo-ecological data about bryozoans, covering in details all aspects of the research done and the future perspectives.

In my opinion the paper is of some interest for all researchers working on marine natural products.

However it is too long and in some parts is unnecessarily redundant.

Some sections appear to me useless, for instance the introduction of section 3, section 7 and section 12.1.

Section 5 is also very repetitive and the author should reduce it.

Two relevant reviews on the topic were not included in the reference section:

Hale, K. J.; Hummersone, M. G.; Manaviazar, S.; Frigerio, M. Nat. Prod. Rep. 2002, 19, 413.

Tian, X.R.; Tang, H.F.; Tian, X.L.; Hu, J.J.; Huang, L.L.; Gustafson, K.R. Review of bioactive secondary metabolites from marine bryozoans in the progress of new drugs discovery. Future Med. Chem. 2018,10, 1497–1514.

There are a huge number of mistakes and inconsistencies.

Some of them are listed below:

Table 1 Line: miscellaneous alkaloids there is not a referred Figure

Figure 3 is missing.

Figure 4: the structures of briostatins 9, 15 and 21 are wrong!!!!!

Two sphingolipids were not reported: Tian XR, Gao YQ, Tian XL et al. New cytotoxic secondary metabolites from marine bryozoan Cryptosula pallasiana. Mar. Drugs 15(4), 120 (2017).

Sentences in Lines 653-669 are the same of 108-124 and are out of the context.

In Figure 7 is not clear the numbering of tambjamine derivatives: for instance why tambjamide J was named 11j or why 10n is missing

Line 1487 please revise the sentence: Kamano et al. in prep in Hashima et al. [423]. Hashima et al. [423]

Line 1557 revise the sentence Total syntheses of wilsoniamine A (46a) and B (46b) were recently proposed by Khan and Ahmad [426], since the paper doesn’t report the synthesis of these compounds.

The text in Figure 19 is not homogeneous.

In figure 20 the names of the schizols A-F doesn’t correspond to those of the original paper.

Author Response

Dear Reviewer,

Here attached is the new version of our review (MS #731209) entitled “The phylum Bryozoa: from biology to biomedical potential, which was previously entitled “Bioactive compounds from bryozoans against cancer, viral and brain and parasitic diseases”, by Maria Letizia Ciavatta, Florence Lefranc, Leandro M. Vieira, Robert Kiss, Marianna Carbone, Willem van Otterlo, Nicole B. Lopanik and Andrea Waeschenbach.

Our revised version has been modified according to each Reviewer’s comment as detailed below.

Reviewer 3:

The present review offers a very comprehensive report of all chemo-ecological data about bryozoans, covering in details all aspects of the research done and the future perspectives. In my opinion the paper is of some interest for all researchers working on marine natural products.

Our response: we warmly thank Reviewer 3 for this positive comment.

However, it is too long and in some parts is unnecessarily redundant.

Some sections appear to me useless, for instance the introduction of section 3, section 7 and section 12.1.

Our response: as for our response to both Reviewer 1 and Reviewer 2, we deleted 80% of the sections relating to cancer in the revised version of our review, and in so doing we thus reduced by more than 60% the length of our review. We also removed from the revised version all the sections relating to PKCs, as also requested by Reviewer 1.

Section 5 is also very repetitive and the author should reduce it.

Our response: section 5 has been deleted from the revised version of our review.

Two relevant reviews on the topic were not included in the reference section:

Hale, K. J.; Hummersone, M. G.; Manaviazar, S.; Frigerio, M. Nat. Prod. Rep. 2002, 19, 413.

Tian, X.R.; Tang, H.F.; Tian, X.L.; Hu, J.J.; Huang, L.L.; Gustafson, K.R. Review of bioactive secondary metabolites from marine bryozoans in the progress of new drugs discovery. Future Med. Chem. 2018,10, 1497–1514.

Our response: we have added Tian et al. in the revised version of our manuscript and also a second review more recent than the one by Hale et al. in 2002, which is old from about 20 years.

There are a huge number of mistakes and inconsistencies.

Some of them are listed below:

Table 1 Line: miscellaneous alkaloids there is not a referred Figure

Our response: Table 1 and the relating “miscellaneous alkaloids” section have been deleted from the revised version of our review.

Figure 3 is missing.

Our response: Fig. 3 is provided in the revised version of our review.

Figure 4: the structures of bryostatins 9, 15 and 21 are wrong!!!!!

Our response: the structures of these bryostatins have been corrected.

Two sphingolipids were not reported: Tian XR, Gao YQ, Tian XL et al. New cytotoxic secondary metabolites from marine bryozoan Cryptosula pallasiana. Mar. Drugs 15(4), 120 (2017).

Our response: most of the sections relating to cancer have been deleted from the revised version.

Sentences in Lines 653-669 are the same of 108-124 and are out of the context.

Lines 653-669 need deleting.

Our response: the section that covered lines 653-669 has been deleted from the revised version.

In Figure 7 is not clear the numbering of tambjamine derivatives: for instance why tambjamide J was named 11j or why 10n is missing

Our response: the figure has been corrected.

Line 1487 please revise the sentence: Kamano et al. in prep in Hashima et al. [423]. Hashima et al. [423]

Our response: the section that covered lines 653-669 has been deleted from the revised version.

Line 1557 revise the sentence Total syntheses of wilsoniamine A (46a) and B (46b) were recently proposed by Khan and Ahmad [426], since the paper doesn’t report the synthesis of these compounds.

The text in Figure 19 is not homogeneous.

In figure 20 the names of the schizols A-F doesn’t correspond to those of the original paper.

Our response: all these parts have been deleted from the revised version.

Hoping that the re-revised version of our review will be acceptable for you, I remain,

Sincerely Yours,

Florence Lefranc, MD, PhD

Reviewer 4 Report

The present review covers literature data dealing with the isolation of bioactive compounds from marine bryozoa against cancer, viral and brain and parasitic diseases.

The manuscript is in general well written with an adequate use of the English language. Moreover, the authors conducted an extensive effort to obtain the information. This is a potentially quite useful review if my two main concerns are addressed:

  1. The chemistry of the bryozoans has been extensively reviewed, including a recent very well-organized review by Tian et al. (2018) (Review of bioactive secondary metabolites from marine bryozoans in the progress of new drugs discovery). I was surprised the authors do not mention this recent review on the same subject. Tian et al. summarize 164 new secondary metabolites including macrocyclic lactones, sterols, alkaloids, sphingolipid. Moreover, another review published last year (Figuerola and Avila 2019) was particularly focused on anticancer drugs isolated from bryozoans as well. Both reviews gave particular attention to anticancer drugs. Another example is a recent review by Wu et al. 2020 (Unlocking the Drug Potential of the Bryostatin Family: Recent Advances in Product Synthesis and Biomedical Applications) that summarizes the recent progress (2014-present) in the development of bryostatins, including their total synthesis and biomedical applications. Therefore, this work did not add much more to our current knowledge on anticancer drugs than both reviews and other ones, where bryostatins were extensively reviewed (there are more examples: Ruan and Zhu 2012). Finally, some papers mentioned are cytotoxic. The authors need to differentiate antitumor from cytotoxicity (cytotoxics are simply inhibitory to cancer cells and this activity does not reflect antitumor activities; e.g. D.J. Newmann in NPR discusses these terms).
  2. Some sections are too long, and many detailed parts are not related to bryozoans or are related to phylogeny. For example, the authors go in depth in marine-derived anticancer drugs from other marine invertebrates. I suggest mentioning only some of them and add the references of the reviews. In addition, some parts are repetitive (see some comments below).

Based on my previous comments, I recommend deleting the anticancer drugs section and focuses on viral and brain and parasitic diseases, which have not been extensively reviewed yet. In particular, antiviral and antiparasitoidal activities sections are poorly developed. The authors should develop them properly as the suggested title mentions them. Alternatively, the title should be more general. Moreover, I recommend shortening other sections. Until these two concerns are not addressed, the rest of the manuscript cannot be evaluated properly although I added some general comments below to reduce a bit the text.

General comments

Section 2.2: this section is too long. It is clear that this part was prepared more from a biological viewpoint and we need a well-constructed phylogeny to guide the search of novel bioactive compounds from bryozoans, however the chemistry is the point of the review and the section can be shortened. The authors could just focus on the current consensus concerning within- bryozoan interrelationships. Also, I can not see the figure 3 along the manuscript. Some suggestions below:

Lines 150-155: Please delete the following sentences from “ Placing bryozoans…” to “…often weak”.

Lines 162-166: Please delete this text from here and add it to the figure legend.

Lines 279-281: Please delete the following sentences from “At the time…”  to “… 1,000 bryozoan species” since it does not provide information for the review.

Lines 287-295: I suggest deleting these sentences from here and adding the information along the manuscript.

Line 317: Please delete “specialized structures where embryos brood until release” as it is already described in a previous section.

Line 440: Please replace the following sentence “see Table 2 for a non-exhaustive list of anticancer marine-derived compounds” with references (e.g. Newman and Cragg) and delete Table 2 as this list is already added in previous recent reviews and papers. This will also reduce a bit the text of the current review.

Line 441: Please delete “ a group of aquatic colonial invertebrate animals” as it is already written before.

Lines 455-456: Please delete the following sentence as it is not necessary: “Bryozoan-related anticancer compounds under review are numbered according to their chronological appearance in the text and listed in Table 1 according to the bryozoan species to which they belong.”

Lines 473- 479: Most text is not related to bryozoans. I suggest adding “(Deng et al, Song et al and Takebe et al)” after “….to combat CSCs” (Line 473) and deleting the sentences from “Deng et al…. to “ …to combat CSCs” (Lines 473- 479).

Section 3.1.6: this section is too long and many detailed parts are not related to bryozoans: the authors go in depth in marine-derived anticancer drugs from other marine invertebrates. I suggest mentioning only some of them and add the references of the reviews.

Lines 569-574: Please delete this text as it is not necessary.

Lines 586-605: I suggest deleting these sentences as the authors describe marine-derived anticancer drugs from other marine invertebrates and these well-known anticancer drugs are described in many other reviews.

Lines 611-617: this part is not linked to previous ones. Please, delete it since it does not say anything about anticancer drugs from bryozoans.

Section 4

Lines 646-669: I do not understand why the authors add these sentences here. Moreover they are already written in a previous section (lines 98-124). Please delete it from here.

Lines 1815-182: Please delete these sentences as this method is widely known for chemists.

Lines 618-1414, 1423-1537, 1551-1558, 1611-1625, 1649-1661, 1892-1909, 1910-1933, 1934-1960: As I already said, this information is already well developed in recent reviews in the same topic. I suggest deleting this part and focuses on viral and brain and parasitic diseases, which have not been extensively reviewed yet and are poorly developed, and other activities (e.g. antibacterial).

Author Response

Dear Reviewer,

Here attached is the new version of our review (MS #731209) entitled “The phylum Bryozoa: from biology to biomedical potential, which was previously entitled “Bioactive compounds from bryozoans against cancer, viral and brain and parasitic diseases”, by Maria Letizia Ciavatta, Florence Lefranc, Leandro M. Vieira, Robert Kiss, Marianna Carbone, Willem van Otterlo, Nicole B. Lopanik and Andrea Waeschenbach.

Our revised version has been modified according to each Reviewer’s comment as detailed below.

Reviewer 4:

The present review covers literature data dealing with the isolation of bioactive compounds from marine Bryozoa against cancer, viral and brain and parasitic diseases.

The manuscript is in general well written with an adequate use of the English language. Moreover, the authors conducted an extensive effort to obtain the information.

Our response: we warmly thank Reviewer 4 for this positive comment.

This is a potentially quite useful review if my two main concerns are addressed:

  • The chemistry of the bryozoans has been extensively reviewed, including a recent very well-organized review by Tian et al. (2018) (Review of bioactive secondary metabolites from marine bryozoans in the progress of new drugs discovery). I was surprised the authors do not mention this recent review on the same subject. Tian et al. summarize 164 new secondary metabolites including macrocyclic lactones, sterols, alkaloids, sphingolipid. Moreover, another review published last year (Figuerola and Avila 2019) was particularly focused on anticancer drugs isolated from bryozoans as well. Both reviews gave particular attention to anticancer drugs. Another example is a recent review by Wu et al. 2020 (Unlocking the Drug Potential of the Bryostatin Family: Recent Advances in Product Synthesis and Biomedical Applications) that summarizes the recent progress (2014-present) in the development of bryostatins, including their total synthesis and biomedical applications. Therefore, this work did not add much more to our current knowledge on anticancer drugs than both reviews and other ones, where bryostatins were extensively reviewed (there are more examples: Ruan and Zhu 2012). Finally, some papers mentioned are cytotoxic. Our response: this comment made by Reviewer 4 is similar to the comment made by Reviewer 2. We thus provide here the same response to Reviewer 4 as the one we provided to Reviewer 2. We thus deleted 80% of the sections relating to cancer in the revised version of our review. We have thus reduced by more than 50% the size of our review. For the cancer topics, we accordingly now refer to the reviews published by Tian et al., Wu et al., and Figuerola and Avila). We have also removed from the revised version all the sections relating to PKCs, as also requested by Reviewer 1.The authors need to differentiate antitumor from cytotoxicity (cytotoxics are simply inhibitory to cancer cells and this activity does not reflect antitumor activities; e.g. D.J. Newmann in NPR discusses these terms).Our response: we are very proud that Reviewer 4 cites the article published by “Dave” Newman in NPR in 2019. It is indeed two of the co-authors of our current review (Florence Lefranc and Robert Kiss) who wrote section 3.1. in this “Newman-related NPR article” about the crucial difference to be made between “cytotoxic and antitumor” activity. Thus, while we have reduced by 80% the cancer topics in the revised version of our review, we nevertheless kept a section that were not covered by the reviews cited by Reviewer 4 and that emphasizes the actual anticancer potential, not the cytotoxic ones, of some bryozoan metabolites. We thus warmly thank Reviewer 4 for having providing us with the great opportunity in improving this part of our review.
    1.  
    2.  
    3.  
    4.  
  • Some sections are too long, and many detailed parts are not related to bryozoans or are related to phylogeny. For example, the authors go in depth in marine-derived anticancer drugs from other marine invertebrates. I suggest mentioning only some of them and add the references of the reviews. Our response: as indicated in our previous comments we markedly reduced the cancer topics in the revised version of our review. In contrast, we would be more than happy if we could convince Reviewer 4 that the phylogeny part provides an important angle for future work, and adds a unique component to our review. This part could be important for other researchers who will be involved in related biomedical research. We nevertheless reduced this section as requested by Reviewer 4.
    1.  
    2.  
  • In addition, some parts are repetitive (see some comments below).     Section 2.2: this section is too long. It is clear that this part was prepared more from a biological viewpoint and we need a well-constructed phylogeny to guide the search of novel bioactive compounds from bryozoans, however the chemistry is the point of the review and the section can be shortened. The authors could just focus on the current consensus concerning within- bryozoan interrelationships. Our response: this section has been shortened as requested by Reviewer 4.Also, I cannot see the figure 3 along the manuscript. Our response: we checked that Fig. 3 has been provided along with the revised version of our manuscript.Some suggestions below:                            Sincerely Yours, Florence Lefranc, MD, PhD
  1.  
  2.  
  3. Hoping that the re-revised version of our review will be acceptable for you, I remain,
  4. Our response: we already responded above to these criticisms raised by Reviewer 4.
  5. Lines 618-1414, 1423-1537, 1551-1558, 1611-1625, 1649-1661, 1892-1909, 1910-1933, 1934-1960: As I already said, this information is already well developed in recent reviews in the same topic. I suggest deleting this part and focuses on viral and brain and parasitic diseases, which have not been extensively reviewed yet and are poorly developed, and other activities (e.g. antibacterial).
  6. Our response: we disagree with Reviewer 4’s comment because while these sentences are indeed widely known for chemists it is not the case for biologists or pharmacologists who represent an important proportion of Marine Drugs’ readers. We thus hope that Reviewer 4 will accept in keeping these sentences in the revised version of our review.
  7. Lines 1815-182: Please delete these sentences as this method is widely known for chemists.
  8. Our response: we did it in the revised version of our review.
  9. Lines 646-669: I do not understand why the authors add these sentences here. Moreover, they are already written in a previous section (lines 98-124). Please delete it from here.
  10. Our response: we did it in the revised version of our review.
  11. Lines 611-617: this part is not linked to previous ones. Please, delete it since it does not say anything about anticancer drugs from bryozoans.
  12. Our response: this section has been deleted in the revised version.
  13. Lines 586-605: I suggest deleting these sentences as the authors describe marine-derived anticancer drugs from other marine invertebrates and these well-known anticancer drugs are described in many other reviews.
  14. Our response: we did it in the revised version of our review.
  15. Lines 569-574: Please delete this text as it is not necessary.
  16. Our response: this section has been deleted in the revised version.
  17. Section 3.1.6: this section is too long and many detailed parts are not related to bryozoans: the authors go in depth in marine-derived anticancer drugs from other marine invertebrates. I suggest mentioning only some of them and add the references of the reviews.
  18. Our response: we did it in the revised version of our review.
  19. Lines 473- 479: Most text is not related to bryozoans. I suggest adding “(Deng et al, Song et al and Takebe et al)” after “….to combat CSCs” (Line 473) and deleting the sentences from “Deng et al…. to “…to combat CSCs” (Lines 473- 479).
  20. Our response: Table 1 and its related text have been deleted from the revised version of our review.
  21. Lines 455-456: Please delete the following sentence as it is not necessary: “Bryozoan-related anticancer compounds under review are numbered according to their chronological appearance in the text and listed in Table 1 according to the bryozoan species to which they belong.”
  22. Our response: we did it in the revised version of our review.
  23. Line 441: Please delete “a group of aquatic colonial invertebrate animals” as it is already written before.
  24. Our response: Table 2 and its related text have been deleted from the revised version of our review.
  25. Line 440: Please replace the following sentence “see Table 2 for a non-exhaustive list of anticancer marine-derived compounds” with references (e.g. Newman and Cragg) and delete Table 2 as this list is already added in previous recent reviews and papers. This will also reduce a bit the text of the current review.
  26. Our response: we sincerely hope that we satisfactorily responded to Reviewer 4 while shortening this section.
  27. Line 317: Please delete “specialized structures where embryos brood until release” as it is already described in a previous section.
  28. Lines 287-295: I suggest deleting these sentences from here and adding the information along the manuscript.
  29. Lines 279-281: Please delete the following sentences from “At the time…”to “… 1,000 bryozoan species” since it does not provide information for the review.
  30. Lines 162-166: Please delete this text from here and add it to the figure legend.
  31. Lines 150-155: Please delete the following sentences from “Placing bryozoans…” to “…often weak”.
  32.  
  33.  
  34.  
  35.  
  36. General comments
  37. Moreover, I recommend shortening other sections. Until these two concerns are not addressed, the rest of the manuscript cannot be evaluated properly although I added some general comments below to reduce a bit the text.
  38. Our response: we have modified the title of our review as requested by Reviewer 4.
  39. Alternatively, the title should be more general.
  40. Our response: as indicated in our response to Reviewer 1, we have indeed deleted 80% of the sections relating to cancer in the revised version of our review. We thus reduced by more than 50% the size of or review as requested by Reviewer 2. For the cancer topics, we now refer to the reviews published by Tian et al. (2018), Wu et al. (2019) and Figuerola and Avila (2020). We did not further develop the remaining topics (not cancer related) because the other Reviewers already and heavily criticized the length of our review.
  41. Based on my previous comments, I recommend deleting the anticancer drugs section and focuses on viral and brain and parasitic diseases, which have not been extensively reviewed yet. In particular, antiviral and antiparasitoidal activities sections are poorly developed. The authors should develop them properly as the suggested title mentions them.

Round 2

Reviewer 1 Report

Please see attached comments.

Author Response

Dear Editor,

Here attached is the revised version (in track mode changes) of our review (MS #731209) entitled “The phylum Bryozoa: from biology to biomedical potential (by Maria Letizia Ciavatta, Florence Lefranc, Leandro M. Vieira, Robert Kiss, Marianna Carbone, Willem van Otterlo, Nicole B. Lopanik and Andrea Waeschenbach).

Our revised version has been modified according to each Reviewer’s comment as detailed below. The modifications we brought in the revised version of our review are indicated here below in bold red while the comments of the Reviewers appear in black italic.

It is our understanding that Reviewer 1 and Reviewer 4 only requested additional modifications for which we hopefully responded satisfactorily as detailed below.

We did not find any additional request from Reviewer 2 and Reviewer 3.

Our responses to Reviewer 1 and Reviewer 4.

Authors have improved their review according to the referees' comments and they did a good job of reducing repetitive parts or parts not related to bryozoans. However, there are still some sections that were already well developed in previous reviews. Therefore, the authors should delete and reduce some parts (see comments below). Also, the text is still full of errors and typos and incorrect references. I have exemplary made corrections along the manuscript but the authors should carefully check the whole manuscript.

We warmly thank Reviewer 4 for these valued comments about our review.

46: Please replace “phyla” with “phylum”

Corrected.

Line 47: Please add the sentence “…and previous reviews in these series” after reference 9 and delete it from the reference list.

The sentence has been added inside the square brackets and deleted in the ref list. Reference 9 has been also updated to that of NPR 2020.

Line 50: Please delete “and” before “illustrating”

Corrected.

Line 52: Please delete “as anticancer”; the verb is lacking: recently reviewed?

Corrected.

Line 53: Please please add “selected” before “compounds”

Corrected.

Line 61:Please replace “bryooan” with “bryozoan””

Corrected.

Line 126: Please replace “monphyletic” with “monophyletic”.

Corrected.

Line 200: Please delete “)”

Corrected.

Lines 254-257: Please delete this sentence and Figure 5 as it is not related to active compounds.

This sentence has been eliminated as well as Figure 5.

Line 285: Please replace with “(5a-n; Fig. 6)”

Corrected.

Line 369: Please add “Cancer Stem Cells (CSCs)” and delete “(CSCs)” from the subtitle (line 467).

Corrected.

Line 429: Please replace “relate” with “related”

Corrected.

Line 447: species name missing

Corrected.

Lines 455 and 457: Please replace “Fig.11” with “Figure 11”

Corrected.

Lines 458-460: Please move these sentences to the figure caption.

Corrected.

Lines 462-463: Please replace “in Fig. 11, which indicates that perfragilin B (10b) displays a mean GI50 concentration of ~4 μM. However, perfragilin B (10b) does not behave as a non selective toxic compound against this panel of 60 cancer cell lines.” with

“Although perfragilin B (10b) displays a mean GI50 concentration of ~4 μM, perfragilin B (10b) does not behave as a non selective toxic compound against this panel of 60 cancer cell lines.”

Corrected.

Lines 464-467: Please replace “Indeed, the bars projecting to the left of the mean GI50 concentration point to those individual cell lines that are less resistant to perfragilin B, while the reverse feature is illustrated by the bars projecting to the right (the most resistant cell lines). Fig. 11 thus shows that perfragilin B (10b) is a rather selective compound, which exerts higher growth inhibitory effects in central nervous system and ovarian (and also some renal) cancers than in leukemia.” with

“Indeed, the bars projecting to the left of the mean GI50 concentration in Figure 11 point to those individual cell lines that are less resistant to perfragilin B, while the reverse feature is illustrated by the bars projecting to the right (the most resistant cell lines). Therefore, perfragilin B (10b) is a rather selective compound, which exerts higher growth inhibitory effects in central nervous system and ovarian (and also some renal) cancers than in leukemia.”

Corrected.

Lines 472-74: replace “with a COMPARE coefficient correlation index (CCCi) of 0.6, which remains weak. In other words, this CCCi value of 0.6 is too weak to state that…” with

“with a weak COMPARE coefficient correlation index (0.6). Therefore, it is not possible to state that…”

Corrected.

Lines 483-488: the sentence is too long. Please split it.

Corrected.

Line 494: please replace “metabolits” with “metabolites”

Corrected.

Lines 569-574: Please delete this text as it is not necessary.

This text has been eliminated as well as the corresponding references from the reference list

Lines 777-832: I already said, most information was well developed in recent reviews on the same topic (e.g. page 3, Tian et al. 2018 and other sections from other reviews). Therefore, it does not provide additional information. Please delete from lines 777 to 801 and reduce the text from lines 813 to 822. Also, another reference wrong is 223 (line 734).

This part have been removed and the reference 223 (line 734) has been corrected into ref 222.

Line 758: reference 36 is wrong. Please check the other references along the manuscript. Also correct the year and replace with “Wu et al. (2020)”

Corrected.

Line 839: Please correct the years of the reviews: Tian et al. (2018), Figuerola and Avila (2019) and Wu et al. (2020)

Corrected.

Line 840: Please replace “informations” with “information”

Corrected.

Lines 842:Please replace “metabolite” with “metabolites”

Corrected.

Figures 2 and 3: Please provide images at higher resolutions.

Figures 2 and 3 are provided with higher resolution.

Please replace the figure 4 for another one with only the chemical structure of bryostatin 1. As already said, bryostatins are previously well reviewed and in the current version you are explaining in detail only bryostatin-1 and its activity.

Corrected.

Figure 8: amathamide E and F has de same letter (7e)

As Figure 5 has been eliminated, Figure 8 is now figure 7; amathamide F now is (7f).

Hoping that this revised version of our review will be acceptable for you, I remain,

Sincerely Yours,

Florence Lefranc, MD, PhD

Reviewer 3 Report

The paper was extensively revised according to all referee's comments.

In particular, all my suggestions have been addressed.

In the present form the paper is now suitable for the publication.

Author Response

(The authors gave the same response as above.)

Reviewer 4 Report

The phylum Bryozoa: from biology to biomedical potential

Authors have improved their review according to the referees' comments and they did a good job of reducing repetitive parts or parts not related to bryozoans. However, there are still some sections that were already well developed in previous reviews. Therefore, the authors should delete and reduce some parts (see comments below). Also, the text is still full of errors and typos and incorrect references. I have exemplary made corrections along the manuscript but the authors should carefully check the whole manuscript.

Comments

46: Please replace “phyla” with “phylum”

Line 47: Please add the sentence “…and previous reviews in these series” after reference 9 and delete it from the reference list.

Line 50:Please delete “and” before “illustrating”

Line 52:Please delete “as anticancer”; the verb is lacking: recently reviewed?

Line 53:Please please add “selected” before “compounds”

Line 61:Please replace “bryooan” with “bryozoan””

Line 126: Please replace “monphyletic” with “monophyletic”.

Line 200:Please delete “)”

Lines 254-257: Please delete this sentence and Figure 5 as it is not related to active compounds.

Line 285: Please replace with “(5a-n; Fig. 6)”

Line 369: Please add “Cancer Stem Cells (CSCs)” and delete “(CSCs)” from the subtitle (line 467).

Line 429: Please replace “relate” with “related”

Line 447: species name missing

Lines 455 and 457: Please replace “Fig.11” with “Figure 11”

Lines 458-460: Please move these sentences to the figure caption.

Lines 462-463: Please replace “in Fig. 11, which indicates that perfragilin B (10b) displays a mean GI50 concentration of ~4 μM. However, perfragilin B (10b) does not behave as a non selective toxic compound against this panel of 60 cancer cell lines.” with

“Although perfragilin B (10b) displays a mean GI50 concentration of ~4 μM, perfragilin B (10b) does not behave as a non selective toxic compound against this panel of 60 cancer cell lines.”

Lines 464-467: Please replace “Indeed, the bars projecting to the left of the mean GI50 concentration point to those individual cell lines that are less resistant to perfragilin B, while the reverse feature is illustrated by the bars projecting to the right (the most resistant cell lines). Fig. 11 thus shows that perfragilin B (10b) is a rather selective compound, which exerts higher growth inhibitory effects in central nervous system and ovarian (and also some renal) cancers than in leukemia.” with

“Indeed, the bars projecting to the left of the mean GI50 concentration in Figure 11 point to those individual cell lines that are less resistant to perfragilin B, while the reverse feature is illustrated by the bars projecting to the right (the most resistant cell lines). Therefore, perfragilin B (10b) is a rather selective compound, which exerts higher growth inhibitory effects in central nervous system and ovarian (and also some renal) cancers than in leukemia.”

Lines 472-74: replace “with a COMPARE coefficient correlation index (CCCi) of 0.6, which remains weak. In other words, this CCCi value of 0.6 is too weak to state that…” with

“with a weak COMPARE coefficient correlation index (0.6). Therefore, it is not possible to state that…”

Lines 483-488: the sentence is too long. Please split it.

Line 494: please replace “metabolits” with “metabolites”

Lines 569-574: Please delete this text as it is not necessary.

Lines 777-832: I already said, most information was well developed in recent reviews on the same topic (e.g. page 3, Tian et al. 2018 and other sections from other reviews). Therefore, it does not provide additional information. Please delete from lines 777 to 801 and reduce the text from lines 813 to 822. Also, another reference wrong is 223 (line 734).

Line 758: reference 36 is wrong. Please check the other references along the manuscript. Also correct the year and replace with “Wu et al. (2020)”

Line 839: Please correct the years of the reviews: Tian et al. (2018), Figuerola and Avila (2019) and Wu et al. (2020)

Line 840: Please replace “informations” with “information”

Lines 842:Please replace “metabolite” with “metabolites”

Figures

Figures 2 and 3: Please provide images at higher resolutions.

Please replace the figure 4 for another one with only the chemical structure of bryostatin 1. As already said, bryostatins are previously well reviewed and in the current version you are explaining in detail only bryostatin-1 and its activity.

Figure 8: amathamide E and F has de same letter (7e)

Author Response

(The authors gave the same response as above.)

Round 3

Reviewer 4 Report

Authors have modified their review according to the referees’ comments. However, the text still has errors and typos. I have made corrections along the manuscript again although they should carefully check the whole manuscript in order to be sure the next version is correct.

Line 45: Please replace “phylumum” with “phylum”.

Line 50: Please replace “illustrateses” with “illustrates”.

Line 51: Please replace “bryozoans metabolites” with “bryozoan metabolites”.

 Line 268: Please replace “A-G.” with “A-G” and delete “7” before “[86-88]”.

Line 285: Please replace “shownn” with “shown”.

Line 391: Please replace “Weran” with “We ran”.

Line 407: Please replace “Although,” with “Although”.

Line 408: μMit?

Line 409: Please replace “barbarIndeed” with “Indeed”.

Line 413: Please replace “cancerscamcers” ” with “cancers”.

Line 421: Please replace “weakCOMPARE” ” with “weak COMPARE”.

Line 422: Please replace “possiblepossible” with “possible”.

Line 434: Please split “caracemideactually”.

Line 533: Please replace “passagepassage” with “passage”.

Author Response

Dear Editor,

Here attached is the revised version (in track mode changes) of our review (MS #731209) entitled “The phylum Bryozoa: from biology to biomedical potential (by Maria Letizia Ciavatta, Florence Lefranc, Leandro M. Vieira, Robert Kiss, Marianna Carbone, Willem van Otterlo, Nicole B. Lopanik and Andrea Waeschenbach).

Our revised version has been modified according to each of the request of Reviewer 4. Indeed, we accordingly corrected all the typo errors as evidenced in the revised version provided in Track mode changes.

We do not resubmit the figures because Reviewer 4 requested no modifications.

Hoping that this revised version of our review will be acceptable for you, I remain,

Sincerely Yours,

Florence Lefranc, MD, PhD